

# Seasonal and interannual variations in the landfast ice mass balance between 2009 and 2018 in Prydz Bay, East Antarctica

Na Li[1], Ruibo Lei[1*], Petra Heil[2,3], Bin Cheng[4], Minghu Ding[5], Zhongxiang Tian[6], Bingrui Li[1]

[1]Key Laboratory of Polar Science of the MNR, Polar Research Institute of China, Shanghai 200136, China
[2]Australian Antarctic Division, Hobart 7001, Australia
[3]Australian Antarctic Program Partnership, University of Tasmania, Hobart 7001, Australia
[4]Finnish Meteorological Institute, Helsinki 00101, Finland
[5]Chinese Academy of Meteorological Sciences, Beijing 100081, China
[6]National Marine Environmental Forecasting Center of the MNR, Beijing 100081, China

*Corresponding author*: Ruibo Lei (leiruibo@pric.org.cn)

**Abstract.** Landfast ice (LFI) plays an important role in the climate and ecosystem of the Antarctic coastal regions. We investigate the LFI snow and ice mass balance in Prydz Bay using data collected by 11 sea ice mass balance buoys (IMBs). The observations were distributed offshore from the Chinese Zhongshan Station (ZS) and Australian Davis Station (DS), and covered 2009–2010, 2013–2016 and 2018 ice seasons. The observed LFI annual maximum ice thickness and snow depth were

1.59±0.17 and 0.11–0.76 m off ZS and 1.64±0.08 and 0.11–0.38 m off DS, respectively. Early in the ice growth season, the LFI basal growth rate near DS (0.6±0.2 cm d$^{-1}$) exceeds that around ZS (0.5±0.2 cm d$^{-1}$). This is attributed to cooler air temperature and lower oceanic heat flux at that time in the DS region. Snow ice contributes up to 27% of the LFI total ice thickness at the offshore site close to ground icebergs off ZS because of the substantial snow accumulation. Larger interannual and local spatial variabilities for the seasonality of LFI mass balance identified at ZS than at DS are due to local differences in

topography and katabatic wind regime. Air temperature anomalies are more important in regulating the LFI growth rate in the early ice growth season because of thinner sea ice relative to later seasons due to the weak thermal inertia of thin ice. Offshore from ZS, the year-round supercooled water from the nearby Dålk Glacier reduces the oceanic heat flux, promoting the LFI growth at the associated sites throughout the entire ice growth season. During late austral spring and summer, we found the increased oceanic heat flux leading to a reduction of LFI growth at all investigated sites. At interannual timescale, we found

that variability of LFI properties across the study domain prevailed, over any trend during the recent decades. We argue that an increased understanding of local atmospheric and oceanic conditions, as well as surface morphology and coastal bathymetry, are required to improve the Antarctic LFI modelling at local and regional scale.



## 1 Introduction

Landfast ice (LFI) is one of the predominant features around the Antarctic coastal zone and typically extends in a narrow predominantly zonal band of varying widths from a few meters to several hundreds of kilometres (Giles et al., 2008; Fraser et al., 2012). LFI is classified as first-year ice in most Antarctic regions and may grow to a thickness of ~2 m via purely thermodynamic processes (Fedotov et al., 1998). Although LFI represents a small fraction (4.0–12.8%) of the overall sea ice extent in the Southern Ocean, it contributes approximately 28% to the total sea ice volume because of its relatively large

thickness compared with drifting sea ice; this contribution is particularly large in summer because of its longer ice season (Giles et al., 2008; Fraser et al., 2021). Its direct coupling with, and immediate responses to, atmospheric and oceanic forcing make LFI a sensitive indicator to climate change, especially at local scales (Heil, 2006; Kim et al., 2018; Arndt et al., 2020). The local distribution of LFI, in some areas, can affect the formation and evolution of polynyas in its downwind region (Nihashi and Ohshima, 2015) and mechanically bond and establish vulnerable outer ice shelf margins (Massom et al., 2018). In addition,

LFI plays a critical role in ice-associated ecosystems as a stable habitat for microorganisms (McMinn et al., 2000), a breeding ground for seals and penguins (Massom et al., 2009), and a support ground for transportation logistics at many Antarctic stations (Kim et al., 2018; Zhao et al., 2020). A better understanding of LFI can provide valuable insights into the climate responses of Antarctic coastal systems. The LFI phenological indexes (freezing up date, ice thickness, ice extent, melting onset date, and ice season length) can indicate the comprehensive and synthetic effects of the local climate system and the

interactions between various physical and biogeochemical processes (Aoki, 2017; Massom et al., 2018; Brett et al., 2020; Arndt et al., 2021). These changeable effects are superimposed with constant local environmental variables, such as topography and other factors, restricting and shaping the seasonal and interannual variations in the Antarctic LFI.

The initial growth of LFI is due to heat loss from the ocean to the atmosphere. Thermodynamic processes dominate the seasonal evolution of most LFI until breakup, except on its edges and shore-connected zones, where compression and shear of the LFI

are easily caused by ocean dynamics, such as tides, waves, or interactions with drifting ice or the shore. The heat fluxes at the top and underside of the LFI exert profound influences on the ice growth (Fedotov et al., 1998; Heil, 2006). The anomalous near-surface air temperature is considered to be one of the factors responsible for the variability in the LFI thicknesses (Heil, 2006; Yang et al., 2015). Snow plays a complex and highly variable role in regulating the Antarctic LFI mass balance via its thermal barrier effect, which limits thermodynamic ice growth in winter (Fichefet and Morales Maqueda, 1999); its high albedo,

which prevents sea ice from melting in summer (Yang et al., 2016; Hao et al., 2020); and formations of snow ice (with brine) or superimposed ice (without brine), which promotes ice growth from the top ice surface (Jeffries et al., 2001). LFI breakouts often occur initially under strong wind and/or wave forcing, which are mostly related to storm events, while the penetration of ocean swells and/or tidal forcing affects the LFI breakup later when protective adjacent pack ice has dispersed (Giles et al., 2008; Lu et al., 2008). Coastal topography and grounded icebergs also play significant roles, primarily by providing shelter

and affecting the snow redistribution (Fraser et al., 2012). The solar and oceanic heat gains drive the surface, internal, and basal melting of LFI in summer (Lei et al., 2010; Zhao et al., 2022). Considering the highly variable nature of the LFI around



Antarctica, thoroughly understanding the physical links of the LFI evolution to the external forcing at local and regional scales is crucial to assess how Antarctic coastal systems respond to climate change.

LFI and its physical properties have been investigated based on data from several Antarctic research stations, such as the
McMurdo station in the McMurdo Sound (Kim et al., 2018; Brett et al., 2020), the Neumayer station in the Akta Bay (Arndt et al., 2020), the Signy station in the South Orkneys (Murphy et al., 1995), the Syowa station in the Lützow-Holm Bay (Aoki, 2017), and the stations within the Prydz Bay (Heil, 2006; Lei et al., 2010). Of these, the Chinese Zhongshan station (hereafter referred to as ZS) and the Australian Davis station (hereafter referred to as DS) are the two main bases for LFI monitoring in the Prydz Bay, the third largest bay around the Antarctic continent (Fig. 1a). In situ LFI measurements near DS have been
taken intermittently since the late 1950s. On the basis of these measurements, the oceanic heat flux under LFI off DS was estimated and the response of the LFI to changes in the local atmospheric conditions was identified (Heil et al., 1996; Heil, 2006). The measurements on the LFI around ZS started relatively late in the early 1990s (He et al., 1998). Largely as a result of discontinuous observations associated with logistic difficulties, previous studies on the LFI around ZS primarily used data collected from only 1 or 2 ice seasons (e.g., Lei et al., 2010; Zhao et al., 2017, 2019). The deployments of ice mass balance
buoys (IMBs) permit the continuous monitoring of the sea ice mass balance without expending significant human resources (Richter-Menge et al., 2006; Jackson et al., 2013).

We collected data from 11 IMBs deployed between 2009 and 2018 at a regional scale on LFI off ZS (6 buoys) and DS (5 buoys). The observations cover the 2009–2010, 2013–2016, and 2018 ice seasons and belong to a part of the Antarctic Fast-Ice Network (AFIN) activities (Heil et al., 2011). The objectives of this study are (1) to quantify the seasonality and the
interannual, as well as spatial, variabilities of the LFI snow and ice mass balance near these two stations, (2) to identify critical factors that are responsible for the LFI variabilities at local or regional scales, and (3) to determine the differences in the LFI thermodynamic processes in response to the local climate changes between the two stations.



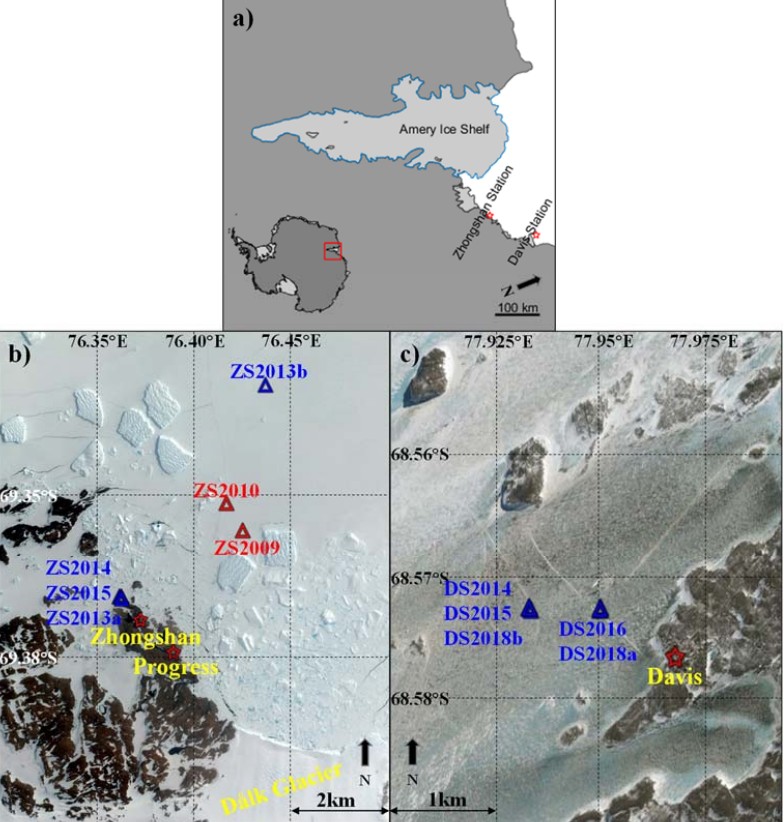

Figure 1. Study area. (a) Close up of the study region in east Antarctica indicated by the red square. The red stars indicate the Chinese Zhongshan station (ZS) and the Australian Davis station (DS). (b) Close up of ZS, where the red triangular symbols indicate the CRREL-IMB deployment sites and the blue triangular symbols indicate the SIMBA deployment sites, with the background showing a Landsat optical image taken on 4 January 2011. (c) Close up of DS, where the blue triangular symbols indicate the SIMBA deployment sites, with the background showing a Landsat optical image taken on 13 October 2012.



## 2 Data and Methods

### 2.1 Observation sites

Located on the eastern rim of the Prydz Bay and affected by the same large-scale climate processes, ZS and DS (Fig. 1a) are approximately 110 km apart and experience similar atmospheric conditions (Streten, 1986). However, the wind forcing at ZS
is characterized by dominant easterly katabatic winds, while the influence of katabatic flow on the climate at DS is less strong (Heil et al., 1996). The topography and bathymetry around ZS are relatively complex, with numerous small islands, grounded icebergs, and undulating submarine topographic features (Feng et al., 2008). The distribution and seasonal evolution of LFI in this region are also affected by the small-scale Dålk Glacier; the width of the front of the Dålk Glacier entering the ocean is approximately 1 km (Chen et al., 2020). Glacial valleys and compressed terrain associated with the Dålk Glacier can strengthen
the katabatic drainage and thus enhance the local wind force. In comparison, the water depths around DS are rather shallow and the offshore islands are sparse. With a similar annual cycle, the LFI at ZS and DS starts to form in early austral autumn (March) and finally breaks up during strong winds and/or high tides in austral summer (from mid-December to late January). The growth and expansion season of LFI in both regions can reach 10 months, which is much longer than its melting season (approximately 2–3 months) (Heil, 2006; Lei et al., 2010). Occasionally, a small portion of LFI may survive through the
summer within small shielded narrow fjords near ZS (Tang et al., 2007; Zhao et al., 2017); however, there is little or no perennial ice observed in the DS LFI record (Heil, 2006); this has been attributed to the differences in the coastal topography between the two regions. Therefore, these two stations are ideal locations to examine the influence of differences in topographical and other local environmental variables on Antarctic LFI processes under similar atmospheric forcing.

### 2.2 IMBs and meteorological data

Two types of IMBs were used for observations: IMBs designed by the U.S. Cold Regions Research and Engineering Laboratory (CRREL-IMB) and Snow and Ice Mass Balance Arrays (SIMBA) designed by the Scottish Association for Marine Science, Scotland. Technical details concerning CRREL-IMB and SIMBA can be found in Richter-Menge et al. (2006) and Jackson et al. (2013). Data collected from two CRREL-IMBs and nine SIMBAs were used in this study. The details concerning the buoy deployments are summarized in Table 1.


Table 1. Details concerning the mass balance buoy (IMB) deployment.



| Buoy | Duration of data record | Deployment position | Water depth (m) | Ice thickness at deployment (m) | Snow depth at deployment (m) | Maximum snow depth (m) | Maximum ice thickness (m) | Date of maximum ice thickness |
|---|---|---|---|---|---|---|---|---|
| ZS2009 | 10 July to 12 December 2009 | 69.36°S 76.42°E | 132 | 1.00 | 0.01 | 0.27 | 1.89 | 24 November 2009 |
| ZS2010 | 26 July 2010 to 31 January 2011 | 69.35°S 76.42°E | 161 | 1.10 | 0.08 | 0.76 | 1.50 | 17 December 2010 |
| ZS2013b | 24 May to 2 October 2013 | 69.33°S 76.44°E | 216 | 0.58 | 0.10 | 0.64[*1] | N/A[*1] | N/A[*1] |
| ZS2013a | 15 May to 27 November 2013 | 69.37°S 76.36°E | ~10 | 0.69 | 0.02 | 0.29 | 1.52 | 22 November 2013 |
| ZS2014 | 13 May to 23 November 2014 | 69.37°S 76.36°E | ~10 | 0.57 | 0.02 | 0.11 | 1.48 | 10 November 2014 |
| ZS2015 | 11 August to 6 December 2015[*2] | 69.37°S 76.36°E | ~10 | 1.16 | 0.18 | 0.44 | 1.54 | 3 December 2015 |
| DS2014 | 29 May to 4 November. 2014 | 68.57°S 77.93°E | ~20 | 0.58 | 0.06 | 0.25 | 1.64 | 15 October 2014 |
| DS2015 | 26 May to 8 December 2015 | 68.57°S 77.93°E | ~20 | 0.88 | 0.10 | 0.25 | 1.60 | 11 November 2015 |
| DS2018b | 22 May to 14 November 2018 | 68.57°S 77.93°E | ~20 | 0.84 | 0.01 | 0.11 | 1.75 | 12 October 2018 |
| DS2016 | 23 May to 27 October 2016 | 68.57°S 77.95°E | 12.4 | 0.80 | 0.06 | 0.38 | 1.54 | 16 October 2016 |
| DS2018a | 28 May to 10 November 2018 | 68.57°S 77.95°E | 12.4 | 0.80 | 0.06 | 0.12 | 1.68 | 31 October 2018 |

Notes: [*1] The annual maximum ice thickness for ZS2013b was unavailable because the buoy was invalidated before the ice thickness reached the annual maximum. The annual maximum snow depth for ZS2013b was obtained from in situ measurements.

[*2] The temperature thermistors of ZS2015 failed to operate properly by 11 August 2015.



The buoys were named according to the deployment location and year. Because of safety concerns, all IMBs were deployed after late April when the ice thickness was greater than 0.50 m. The ZS2009, ZS2010, and ZS2013b deployment sites were approximately 3–6 km northeast of the shore with bathymetries deeper than 100 m (Fig. 1b); this avoids the shallow water and small islands near the shore and can be representative of LFI in the ZS offshore region. The ZS2013a and ZS2014 SIMBAs,

together with ZS2015, were deployed at sites approximately 100 m off the coastline northeast of ZS, with nearshore characteristics. The DS2016 and DS2018a SIMBAs were deployed approximately 1 km northwest off the coast of DS (Fig. 1c); this is a regular LFI measurement site and is defined as S1 in Heil (2006). The DS2014, DS2015, and DS2018b SIMBAs were deployed approximately 1.5 km northwest of DS. All the deployment sites were classified as first-year ice. To ensure that the borehole for the buoy deployment was fully refrozen, our analyses used only data obtained 10 days after depolyment.

The hourly meteorological parameters in the period of 1989–2018, including the near-surface (at altitudes of 18 m at ZS and of 13 m at DS) air pressure (AP), air temperature (AT), relative humidity (RH), and wind speed and direction (WS and WD, respectively), measured at the ZS and DS meteorological stations were used to characterize the local atmospheric forcing. Solid precipitation (SP) was measured intermittently with daily intervals at the Russian Progress II station (approximately 1 km from ZS) and was available in the form of the snow-to-water equivalent (SWE) during the period of 2009–2018 from the

global Integrated Surface Hourly database.

## 2.3 Determinations of the snow depth and ice thickness

For the CRREL-IMBs, the interfaces between the air and the snow or ice (in the absence of snow), as well as between the ice and the water, were measured using acoustic sounders with an accuracy of 0.01 m, from which the snow depth and ice thickness can be derived by assuming that the interface between the snow and the ice remains unchanged. Unfortunately, the data from

the underwater acoustic sounder of ZS2009 were available only from 10 July to 2 August 2009 because of a sensor failure. In the absence of direct measurements, we estimated the ice thickness of ZS2009 based on the difference in the vertical temperature gradients between the ice and the water (e.g., Perovich and Elder, 2001). The estimation error may be up to 0.05 m because of the relatively large vertical interval of the thermistor sensors for this style of IMB. SIMBAs feature a heating mode that provides the temperature rise after pulsed heating (HT) of 30 and 120 s (Jackson et al., 2013). The snow depths and

ice thicknesses for the SIMBAs were derived from the vertical profiles of the temperature and the HT (Provost et al., 2017; Liao et al., 2018). The snow depth and ice thickness were validated using in situ borehole measurements at the buoy deployment sites during the buoy operation period from 2013 to 2015 off ZS and DS, with sampling intervals ranging from 3 days to 1 month. Compared with in situ observations, the temperature-profile-based estimations provided high accuracies of $0.1 \pm 4.4$ cm for the snow depth and $0.6 \pm 3.0$ cm for the ice thickness. Note that the topmost temperature thermistor of ZS2014

was just placed on the snow–ice interface at deployment as a result of an inaccurate operation. In addition, the topmost thermistor of ZS2013b was submerged within the snow by 6 September 2013 because the accumulated snow was too thick.


Because the snow depth could not be retrieved from the temperature profiles at these two sites, the in situ measurements were used instead.

The upper ice surface might shift vertically as a result of the formation of snow ice or superimposed ice (Jeffries et al., 1998).

The ratio between the 30 s and 120 s HTs can be used to discriminate materials with different heat-capacity regimes, especially snow and sea ice (Jackson et al., 2013). Consequently, the data can be used to identify the formation of snow ice or superimposed ice. On the basis of a manual inspection of the vertical profiles of the HT ratio, vertical shifts in the snow–ice interface were clearly observed in the ZS2013a and ZS2013b data (Fig. 2). At other sites with SIMBA deployment, there were no identifiable changes in the snow–ice interface, suggesting the lack of snow ice or superimposed ice formation.


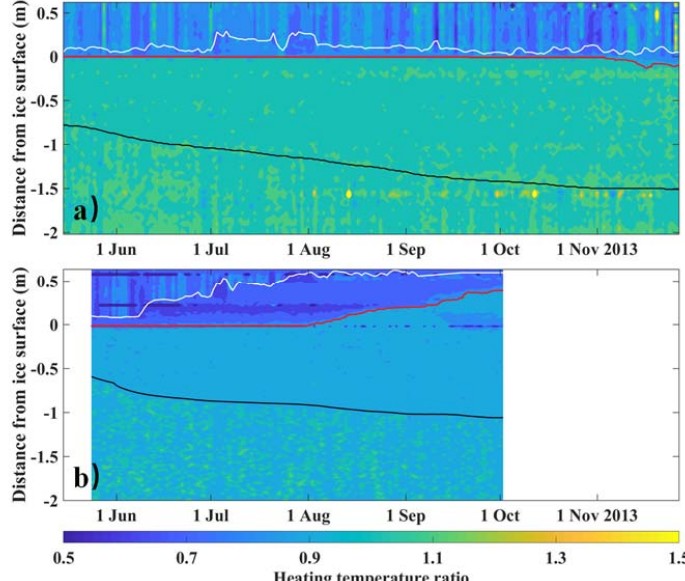

Figure 2. Profiles of the pulsed heating (HT) ratio after 30 and 120 s observed by the (a) ZS2013a and (b) ZS2013b Snow and Ice Mass Balance Arrays (SIMBAs), with the white, red, and black lines denoting the snow surface, the snow–ice interface, and the ice base, respectively. The zero line refers to the initial snow–ice interface when the buoy was deployed.


To identify the probability of the occurrence of snow ice, the ice freeboard, $H_f$, for each IMB was estimated using Archimedes buoyancy principle (Provost et al., 2017):

$$H_f = H_i - \frac{\rho_i H_i + \rho_s H_s}{\rho_w},$$   (1)





where $H_i$ and $H_s$ are the ice and snow thicknesses, respectively, and $\rho_i$, $\rho_s$, and $\rho_w$ are the densities of the sea ice, snow, and
seawater, with typical density values of 900, 330, and 1028 kg m$^{-3}$, respectively (e.g., Lei et al., 2010). A negative ice freeboard
is one of the most important mechanisms for the formation of snow ice (Rösel et al., 2018).

**2.4 Heat fluxes through the ice and at the ice base**

At the ice base, the heat fluxes satisfy the balance:

$$F_c + F_l + F_s - F_w = 0 , \tag{2}$$

where $F_c$ is the conductive heat flux through the basal ice layer, $F_l$ is the equivalent latent heat flux caused by ice freezing or
melting, $F_s$ is the specific heat flux caused by ice warming or cooling, and $F_w$ is the oceanic heat flux. The upward ($F_c$ and $F_w$),
melting ($F_l$), and warming ($F_s$) heat fluxes are defined as positive. $F_l$ and $F_s$ are calculated based on the ice growth/melt rate
and the temporal changes in the ice temperature, respectively. $F_c$ was estimated based on the vertical ice temperature gradient
(Lei et al., 2010). The latent heat, specific heat, and thermal conductivity of the sea ice used to estimate these heat fluxes are
functions of its temperature and salinity (Ono, 1967; Pringle et al., 2006). Here, to calculate $F_c$ and reduce the influence of the
nonlinear vertical temperature gradient in the ice basal skeleton layer, we define the basal layer as the layer 0.12–0.24 m above
the ice base (Lei et al., 2010). In our calculations, we linearly interpolated the temperatures of the CRREL-IMBs to an interval
of 2 cm, consistent with the measurements of the SIMBAs. The density and salinity of the newly formed basal ice layer were
set to 910 kg m$^{-3}$ and 8, respectively. The 15 day moving average data were used for the estimations of the heat fluxes to
reduce the uncertainties in the measurements. We quantified these heat fluxes to characterize their respective contributions to
the sea ice basal growth, as well as the associated seasonal and interannual variations.

The conductive heat flux was also estimated for layers with thicknesses of 0.12 m throughout the entire sea ice column to
assess the delayed response of the sea ice cooling at the lower layer and the ice basal growth with respect to the changes in the
atmospheric forcing. To calculate the conductive heat fluxes, we used a sea ice density of 910 kg m$^{-3}$, and an ice salinity of 3
for the top 0.5 m section (considering its desalination; e.g., Lei et al., 2010) and 5 for the section below 0.5 m.

**2.5 Assessing the effects of the snow cover, AT anomaly, and oceanic heat flux on the LFI growth**

Under cold conditions, a linear vertical temperature change is adequate between the ice top surface at $T_s$ and the base at the
freezing point $T_f$; the thermodynamic ice growth can be approximatively estimated by equating the latent heat removed by the
vertical heat conduction upward through the ice to the colder air above (Stefan, 1891). When considering the contributions of
the oceanic heat flux, the revised Stefan law (Leppäranta, 1993) is given as

$$\frac{dH_i}{dt} = -\frac{k_i}{\rho_i L_i}\frac{\partial T}{\partial z} + \frac{F_w}{\rho_i L_i} = -\frac{k_i}{H_i \rho_i L_i}\left(T_s - T_f\right) + \frac{F_w}{\rho_i L_i} , \tag{3}$$





where $dH_i/dt$ is the ice growth rate, $\rho_i$ is the density of the sea ice, $L_i$ and $k_i$ are the latent heat and thermal conductivity of the sea ice, respectively, and $\partial T/\partial z$ is the vertical temperature gradient in the ice. With an initial ice thickness $H_0$, the analytical solution for the ice thickness $H_i$ can be estimated as

$$H_i = \sqrt{H_0^2 + \alpha^2\theta} - \frac{1}{\rho_i L_i}\int F_w dt \,, \tag{4}$$

where

$$\alpha = \sqrt{2k_{si}/\rho_{si}L_f} \,. \tag{5}$$

Here $\theta$ is the integral of the negative ice surface temperature $T_s$ below the freezing point $T_f$ (here, defined as −1.9 °C) over time.

To identify the impact of snow cover on the LFI mass balance from the perspective of the thermal insulation effect, we used the AT obtained from the year of observation (AT_obs) instead of $T_s$ for the LFI thickness calculation; this leads to $\theta$ being equivalent to the freezing degree days. To identify the influence of the AT anomaly during the buoy operation years on the LFI mass balance, the LFI growth derived using the revised Stefan analytical model forced by AT_obs was compared with that forced by the 1989–2018 average AT (AT_mean). To assess the effect of the oceanic heat flux on the LFI growth, we compared the evolutions of the ice thickness from the beginning of the observations to the end of October estimated using Eq. (4) with those estimated ignoring the oceanic heat flux.

## 3 Results

### 3.1 Atmospheric conditions

Compared with the buoy-observed data over 10 years, the mean biases of the AT measured at the ZS and DS onshore weather stations were very small, with values of 0.4 ± 1.6 and 0.3 ± 2.2 K, respectively. Therefore, we used the meteorological station data for the following analysis of the atmospheric conditions to identify the anomalies in the buoy operation years compared with the climatology. Because the meteorological data from ZS are available since 1989, we used the data from the period of 1989–2018 to identify the anomalies for both stations.

At ZS, during the period of 2009–2018, the annual mean AT was −10.2 °C, with the highest value observed in 2009 (−9.3 °C) and the lowest observed in 2015 (−11.3 °C); this is comparable to the 1989–2018 climatology (−10.0 °C) (Table 2 and Fig. 3a). The AT from May to September averaged for each year was defined as the mean AT of the LFI rapid growing season, with a value of −15.9 °C at ZS during the period of 2009–2018 (Fig. 3a). The annual mean wind speed was 6.6 m s$^{-1}$, 73% of which accounts for wind from the east-northeast–east-southeast direction (Fig. 3d). Under the influence of persistent cold dry katabatic winds, the annual accumulative SWE of the SP was relatively low, with an average of 192.8 ± 59.9 mm during the





period of 2009–2018. The annual SP varied greatly from year to year, with larger amounts of precipitation observed in 2013 (239.3 mm) and 2015 (230.8 mm) and smaller amounts of precipitation observed in 2009 (124.5 mm) and 2018 (153.0 mm). Compared with ZS, DS has colder winters and warmer summers (Fig. 3b). During the period from 2009 to 2018, the winter and summer AT values at DS were 0.7 °C colder and 0.5 °C warmer, respectively, than those at ZS. The difference between the mean AT of the LFI rapid growing season at the two stations (DS − ZS) was −0.6 °C on average. During the period of

2009–2018, the lowest AT occurred in 2015 (−11.4 °C) and the highest AT occurred in 2018 (−9.0 °C) at DS, with large AT anomalies primarily occurring between May and September (Fig. 3b). On the edge of the low-lying Vestfold Hills, DS suffers relatively mild wind conditions, with an annual mean wind speed of 6.1 m s$^{-1}$. The occurrence of a relatively high wind speed larger than 8 m s$^{-1}$, namely the threshold wind speed for drifting snow (Schmidt, 1981), at DS was 23%, which was much smaller than that at ZS (35%). With less influence of dry katabatic winds, the long-term mean of the annual accumulative SWE of the SP at DS was 231 mm (Heil, 2006), approximately 40 mm larger than that at ZS during the period of 2009–2018.

Table 2. Statistics concerning the principal meteorological parameters at the Chinese Zhongshan station (ZS) and the Australian Davis station (DS).

| Station | Meteorological parameters | Average (±Std) in 1989–2018 | Average (±Std) in 2009–2018 | Maximum in 2009–2018 | Minimum in 2009–2018 |
|---|---|---|---|---|---|
| ZS | Annual mean AT (°C) | −10.0±0.9 | −10.2±0.7 | −9.3 (2009) | −11.3 (2015) |
|  | Summer (DJF) mean AT (°C) | −1.0±0.8 | −0.9±0.6 | 0.0 (2016) | −1.9 (2010) |
|  | Winter (JJA) mean AT (°C) | −15.7±1.9 | −16.5±1.8 | −13.3 (2018) | −18.1 (2015) |
|  | Annual mean WS (m s$^{-1}$) | 7.0±0.6 | 6.6±0.6 | 7.5 (2018) | 5.8 (2016) |
|  | Annual SWE (mm) | 158.9±45.6 | 192.8±59.9 | 320.8 (2017) | 120.3 (2012) |
| DS | Annual mean AT (°C) | −10.1±1.0 | −10.2±0.7 | −9.0 (2018) | −11.4 (2015) |
|  | Summer (DJF) mean AT (°C) | −0.3±0.7 | −0.4±0.5 | 0.3 (2016) | −1.2 (2010) |
|  | Winter (JJA) mean AT (°C) | −16.7±2.0 | −17.2±1.8 | −13.4 (2018) | −19.0 (2014) |
|  | Annual mean WS (m s$^{-1}$) | 6.0±0.6 | 6.1±0.2 | 6.4 (2017) | 5.8 (2016) |
|  | Annual SWE (mm) | 231±92[*] | / | / | / |

Note: * Extracted from Heil (2006).

"/" indicates that the data were unavailable.




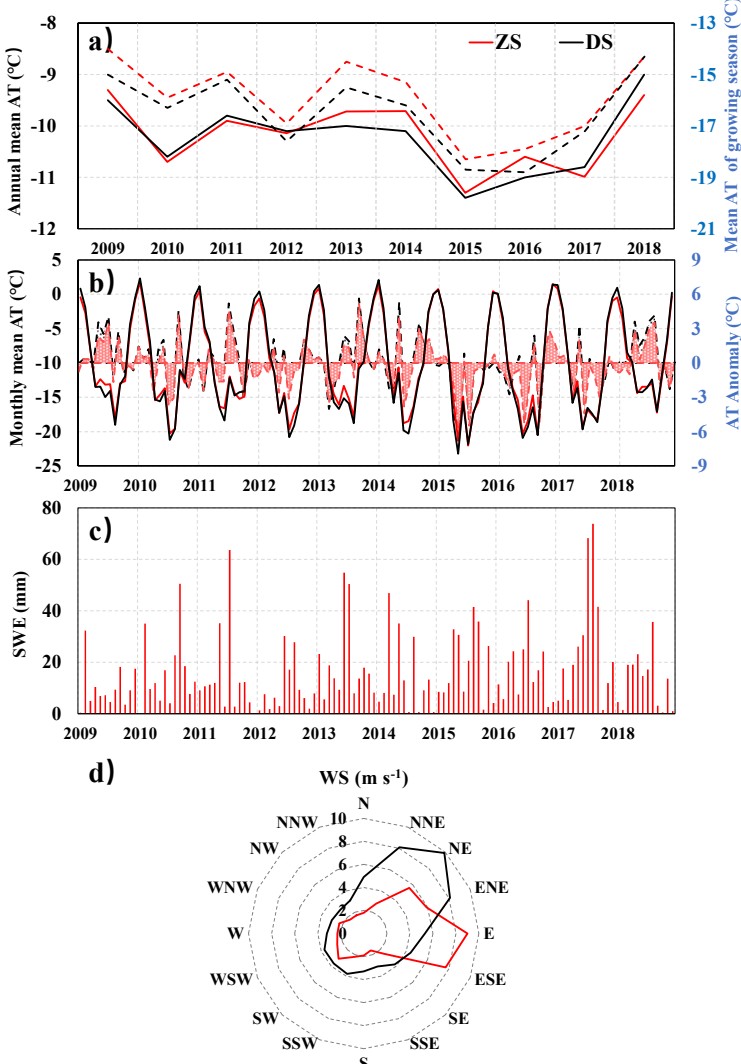

Figure 3. Atmospheric parameters measured by weather stations at ZS (red) and DS (black) from 2009 to 2018: (a) annual mean air temperature (AT; solid line, left axis) and mean AT of the landfast ice (LFI) rapid growing season (from May to September, dashed line, right axis), (b) monthly mean AT (solid line, left axis) and AT anomaly (shaded area, right axis), (c) monthly accumulative snow-to-water equivalent (SWE; obtained from the Russia Progress II station close to ZS), and (d) wind speed (WS) and wind direction (WD) distribution.





### 3.2 Snow and sea ice mass balance at ZS

At the buoy deployments near ZS, the snow cover was relatively thin in mid- and late May (Fig. 4a), with a depth of less than 0.10 m. Most of the high snow accumulation rates were obtained synchronously with large SP amounts measured at the

Progress II station (Fig. 3c). The most noticeable snow accumulation was observed at ZS2010, with the snow depth increasing from 0.40 to 0.73 m from 13 to 15 September 2010. During this period, two snowfall events occurred with the SWE of the SP amount being 26.1 mm, contributing to 0.11 m of on-site snow accumulation. Considering that the ZS2010 buoy was deployed in an area with enriched grounded icebergs (Fig. 1b), the persistent east-northeast wind after the first snowfall event and the sheltering effect of the grounded icebergs and growlers may have promoted snow accumulation on their lee sides and led to

the additional increase in the snow accumulation in these locations. After the second snowfall event, strong winds occurred with speeds reaching 20.6 m s$^{-1}$ and a directional change from east-northeast to east. Therefore, the accumulated snow was prone to being blown away. Consequently, the snow depth sharply decreased to 0.48 m by 17 September 2010. It is clearly shown that blowing snow played a vital role in the snow redistribution, especially for fresh noncompact snow cover, and contributed to the dramatic variations in the snow depth over the LFI off ZS.


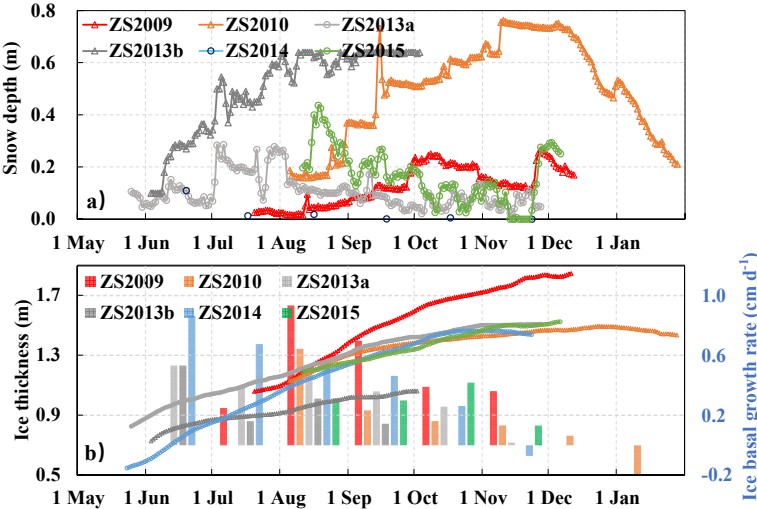

Figure 4. Seasonal evolution of (a) the snow depth and (b) the ice thickness (lines, left axis) off ZS derived from CRREL-IMB and SIMBA observations, with the vertical bars in panel (b) denoting the monthly ice basal growth rates (right axis).

The snow depths at all of the measurement sites reached their annual maximums from late June to late November (Fig. 4a), which is a very long temporal span, indicating that nearly all annual maximum snow depths were related to episodic synoptic



events, not seasonal accumulation. At the inshore sites (ZS2013a, ZS2014, and ZS2015), the annual maximum snow depth ranged between 0.11 and 0.44 m. Meanwhile, at the offshore sites (ZS2009, ZS2010, and ZS2013b), the maximum snow depth was relatively large, with values reaching 0.27 (ZS2009), 0.76 (ZS2010), and 0.64 m (ZS2013b). Compared with the inshore

sites, the relatively weak wind forcing and the shielding effect of the local topography of the small islands and grounded icebergs were likely related to the thicker snow cover at the offshore sites. With the lower occurrence frequency of snowfall in spring, except for occasional cases (e.g., in late November in 2009 and 2015), the snow depth gradually decreased with the combined processes of surface sublimation and snow compaction. When the air temperature rose above 0 °C after December, the snow depth dropped sharply; this can be clearly observed at nearly all buoy deployment sites. As a result of the relatively

thin snow cover in the late ice growing season at ZS2014, ice surface melt occurred starting in mid-November at this site, much earlier than at the other sites.

The growth and expansion of LFI around ZS are not a continuous process. In general, the onset of a continuous LFI cover occurred in early to mid-March (Lei et al., 2010). After repetitive processes of ice formation, breakup, and reformation during the early freezing season as a result of wind forcing and/or ocean dynamics, the LFI cover around ZS became relatively stable

from April onward. Most of our IMB observations started in May. Three growth stages were identified based on the LFI thermodynamic growth process (Fig. 4b). From May to September, the LFI was in a rapid growth stage, with an average monthly ice basal growth rate of $0.5 \pm 0.2$ cm d$^{-1}$. In our buoy dataset, the maximum monthly ice basal growth rate occurred in August 2009, reaching 0.9 cm d$^{-1}$. During the period of October–November, with the thickening of the ice cover and the increase in AT, the ice growth gradually decelerated and ceased, suggesting a steady growth stage with a monthly basal growth

rate of $0.2 \pm 0.2$ cm d$^{-1}$. The annual maximum ice thickness was observed by the end of this stage (i.e., mostly in late November), with a value of $1.59 \pm 0.17$ m obtained from five sites measured in 5 years (Table 1). Note that the annual maximum ice thickness at ZS2013b could not be obtained because the observation was invalidated before the LFI thickness reached its annual maximum. After a state of thermal equilibrium, the LFI near ZS entered a melt stage from late November until the ice breakup occurred in late December or early January (Lei et al., 2010).

The evolution of the LFI mass balance also exhibits large spatial variations. Approximately 6 km apart, the ZS2013a and ZS2013b sites experienced nearly the same atmospheric conditions; however, the total ice basal growths at these two sites revealed large differences, with values of 0.52 and 0.33 m from 3 June to 1 October, equivalent to latent heat fluxes of 10.2 and 6.5 W m$^{-2}$, respectively. Located within a typical distance from the shore (6–15 km) where continental winds diminish and drifting snow tends to accumulate (Fedotov et al., 1998), significant amounts of accumulated snow were observed at

ZS2013b. The snow cover at ZS2013b reached 0.30 m in mid-June, approximately 0.20 m thicker than that at ZS2013a, which efficiently insulated the LFI from the cold atmosphere and substantially reduced its basal growth. However, the snow depth at ZS2013a decreased rapidly and maintained a value of approximately 0.10 m after reaching its annual maximum (0.29 m) in early July. By the end of the observation period in early October 2013, the LFI thickness at ZS2013b was 1.06 m, much smaller than the thickness (1.42 m) at ZS2013a.



### 3.3 Snow and sea ice mass balance at DS

The seasonal evolution of the snow depth at DS remained rather smooth and the amplitude of the short-term fluctuations were much smaller compared with those at ZS because of the relatively open terrain and mild wind conditions (Figs. 5a and 4a). At deployment in late May, the snow depth around DS ranged between 0.01 and 0.10 m. Snow accumulated gradually from May to July. The annual maximum snow depths all occurred in winter (from June to August), with values varying from 0.11 to 0.38 m. From then onward, the snow depth gradually decreased until the end of the measurements in December.

LFI around DS started to appear from early March and formed a stable cover by April. The annual maximum ice thickness was observed between mid-October and mid-November (Table 2), with the mean value obtained from five IMBs being $1.64 \pm 0.08$ m, which was slightly larger than the mean value obtained at ZS ($1.59 \pm 0.17$ m). Entering the melting stage, final LFI breakup was observed from mid-December to February and was usually associated with passing cyclonic activity (Heil, 2006).

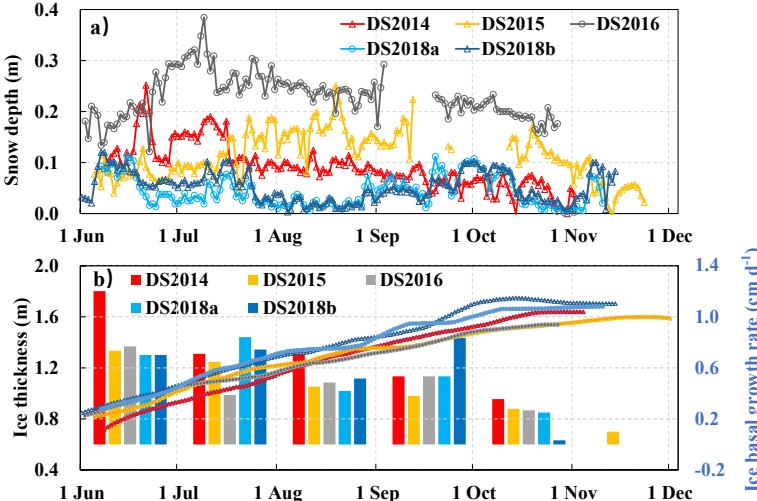

Figure 5. Same as Figure 4, but for DS.

IMB observations were made simultaneously on the LFI near DS and ZS during the two ice seasons of 2014 and 2015. The snow cover did not differ much between the two sites during the same year except for variations related to several synoptic events. However, during the rapid ice growth stage from May to September, the LFI grew faster at DS with an ice basal growth rate of $0.6 \pm 0.2$ cm d$^{-1}$ compared with $0.5 \pm 0.2$ cm d$^{-1}$ at ZS (Figs. 4b and 5b). Correspondingly, the LFI thickness at DS2014 and DS2015 increased to 1.52 m and 1.47 m, respectively, by the end of September; the values were 0.15 m and 0.13 m thicker than those at ZS2014 and ZS2015, respectively. This may be partly attributable to the relatively lower ATs at DS during this



period (Fig. 3a), during which time the freezing degree days at DS were 2189.5 K d and 2572.9 K d in 2014 and 2015, respectively, approximately 6.8% and 2.4% larger than those at ZS, with values of 2049.8 K d and 2513.4 K d, respectively. After September, with the increase in AT and the increase in the ice thickness, the LFI growth at DS entered the steady growth stage; the maximum ice thickness at DS occurred approximately 20 days earlier than that at ZS (Table 1). Because of the relatively warm summer and the lack of shelter from coastal islands and grounded icebergs, the LFI broke up earlier off DS

than off ZS. There was an approximately 1 month ice-free period in the vicinity of DS, which was somewhat longer than that off ZS by approximately 10–20 days (Heil, 2006; Lei et al., 2010).

### 3.4 Sea ice temperature profile and heat flux

As an example, Fig. 6 presents profiles of the snow and sea ice temperatures and the conductive heat flux through the LFI at ZS2013a. The conductive heat fluxes centred at depths of 12 cm below the snow–ice interface and 12 cm above the ice–ocean

interface were used to indicate the surface and bottom layers of the ice column (Fig. 6c), respectively, which participate in the thermal balance with the snow and the ocean. Note that the thin layers (6 cm) near the upper and lower boundaries of the ice cover are not considered here, primarily because these two thin layers have complex texture structures; that is, they are scattering layers similar to the snow at the top layer and the mush layer at the ice bottom.


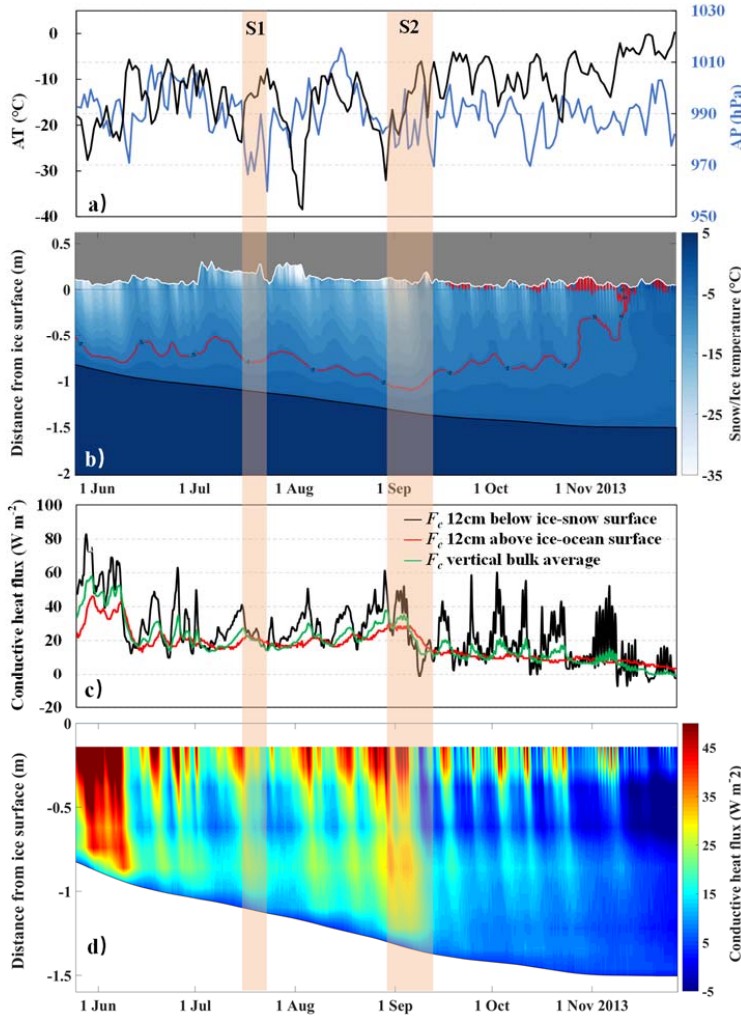


Figure 6. (a) Time series of the AT (black line, left axis) and air pressure (AP, blue line, right axis); (b) snow and sea ice temperatures, with the white, red, and black lines representing the snow surface, the −5 °C isotherm, and the ice base, respectively; (c) conductive heat flux through the ice (black: centred 12 cm below the snow–ice interface, red: centred 12 cm above the ice–ocean interface, and green: the vertical bulk average); and (d) conductive heat flux through the ice at ZS2013a. On the vertical axes, the zero line refers to the initial snow–ice

interface. The shaded areas represent periods experiencing two typical winter warming events, S1 and S2.



During winter, when the study region was under the control of the Antarctic continental cold high atmospheric system, a rapid decrease in the temperatures occurred in the snow cover and upper layers of the ice in response to the decreasing AT (Figs. 6a and 6b). The variation range of the temperature at the snow–ice interface was only half of that of AT, from which the thermal

insulation effect of a snow cover can be readily appreciated. The snow layer, with a smaller density and thermal conductivity, had an obviously larger temperature gradient compared with the ice. Conversely, episodic warming events occurred frequently in winter when low-pressure synoptic systems were in place. During these events (e.g., two typical events, S1 and S2, are shown in Fig. 6a), the temperature gradient in the snow layer became inverted, with the local temperature minimum occurring at the snow–ice interface or the top layer of the ice. With the sharp increase in AT and the thickening of the snow cover during

these events, the conductive heat flux at the ice surface layer dropped dramatically by approximately 20 W m$^{-2}$ within 3–5 days in both cases. The fluctuation of the conductive heat flux near the ice bottom was relatively smooth and revealed a temporal lag relative to that at the ice surface layer, which may be related to the buffering effect of ice layers and the internal brine (Fig. 6c). This temporal lag changed the direction of the vertical heat conduction within the ice column (Fig. 6d). From spring onward, the diurnal signal in the temperature profile became increasingly pronounced at the top ice layers because of

the increase in the incoming solar radiation and the bulk conductive heat flux through the entire ice column decreased to several W m$^{-2}$ (Fig. 6c). After 10 November 2013, negative conductive heat flux occurred in the top approximately 0.7 m ice layer (Figure 6d); this was attributed to the temporal lag of the warming of the lower ice layers relative to the increase in AT as a result of the heat storage capacity of the brine within the ice. At this time, a downward conduction of the heat flux into the middle layer of the ice was observed, promoting further ice interior melting. Therefore, an internal melting gap layer was often

observed approximately 10 cm to tens of centimeters below the ice surface at ZS; this weakened the ice layer and promoted the breaking up of the LFI (Zhao et al., 2022).

To assess the relative contributions of each component of the heat fluxes at the ice bottom to the LFI growth, the basal heat flux components at each buoy site were estimated and compared (Fig. 7). Throughout the entire observation period, because the temperature at the ice base was nearly constant, $F_s$ was always relatively small. The basal ice growth was primarily

regulated by $F_c$ and $F_w$. $F_c$ contributed approximately 50% to the basal ice energy balance (Figs. 7i and 7j). For the nearshore (ZS2013a) and offshore (ZS2013b) sites around ZS (Figs. 7a and 7b), the maximum $F_c$ occurred on 3 June, with a value of 45.2 W m$^{-2}$ at ZS2013b, approximately 8.0 W m$^{-2}$ larger than the value at ZS2013a. At this time, the basal ice growth rate at ZS2013b was 0.7 cm d$^{-1}$, comparable to the rate at ZS2013a; this is primarily attributable to the difference in the oceanic heat fluxes under the ice between two sites. During the initial freezing period with a large ice growth rate, the surface water was

densified by brine rejection from the sea ice; this drove vertical mixing. The coastal water at ZS2013a is shallow (<10 m) and is expected to be entirely mixed. Approximately 6 km offshore, the water depth at ZS2013b is up to 216 m. Therefore, the potential upward entrainment of warmer deep water at this site likely resulted in a higher value of the ocean heat flux (22.2 W m$^{-2}$) and provided a thermal constraint on the LFI growth. After mid-June 2013, $F_w$ at both the ZS2013a and ZS2013b sites dropped to less than 10 W m$^{-2}$. Meanwhile, with the increasing snow depth, as mentioned above, $F_c$ at the ice basal layer of





ZS2013b decreased sharply to approximately half that of ZS2013a. Correspondingly, the basal ice growth rate at ZS2013b decreased to 0.3 cm d$^{-1}$, 60% of the rate at ZS2013a (0.5 cm d$^{-1}$).

When compared, the measurements obtained from the coastal sites near ZS and DS during the same year (ZS2014 versus DS2014, as shown in Figs. 7c and 7d, and ZS2015 versus DS2015, as shown in Figs. 7e and 7f), $F_c$ revealed a similar seasonal pattern for the two sites, with the largest value observed in June and then decreasing gradually until the end of the observations

by late October in 2014 or late November in 2015. The mean $F_c$ values from June to October 2014 and from August to November 2015 were comparable in the same year between the two sites, with values of 17.0 and 10.8 W m$^{-2}$ at ZS and 16.5 and 10.7 W m$^{-2}$ at DS, respectively. Meanwhile, the mean $F_w$ under the LFI from June to September (3.2 W m$^{-2}$), during the stage of rapid ice growth, and the contribution of $F_w$ to the basal ice energy balance (14.1%) at ZS were much larger than those at DS (0.7 W m$^{-2}$ and 8.8%, respectively). This relatively large oceanic heat flux, which reduced the ice basal growth, together

with the relatively small mean $F_c$, led to the thinner annual maximum LFI thickness at ZS than at DS (1.48 m versus 1.64 m in 2014 and 1.54 m versus 1.60 m in 2015). Approximately 500 m apart, DS2018a and DS2018b had nearly the same snow depth and ice thickness evolutions, resulting in similar conductive heat fluxes through the ice (Fig. 7g and 7h). Further away from the shore, the $F_w$ at DS2018b continued to increase from a negative value at the beginning of October to 15 W m$^{-2}$ by mid-October. Combined with the decreased $F_c$, this resulted in the earlier onset (17 October) of ice basal melt at DS2018b than

at DS2018a (3 November). Note that the relative contributions of each component to the total heat balance at the ice bottom shown in Figs. 7i and 7j were obtained from multi-site measurements, which can be considered to be the average state of the LFI heat balance at the two stations. This is valuable for the validation of regional LFI numerical simulations.
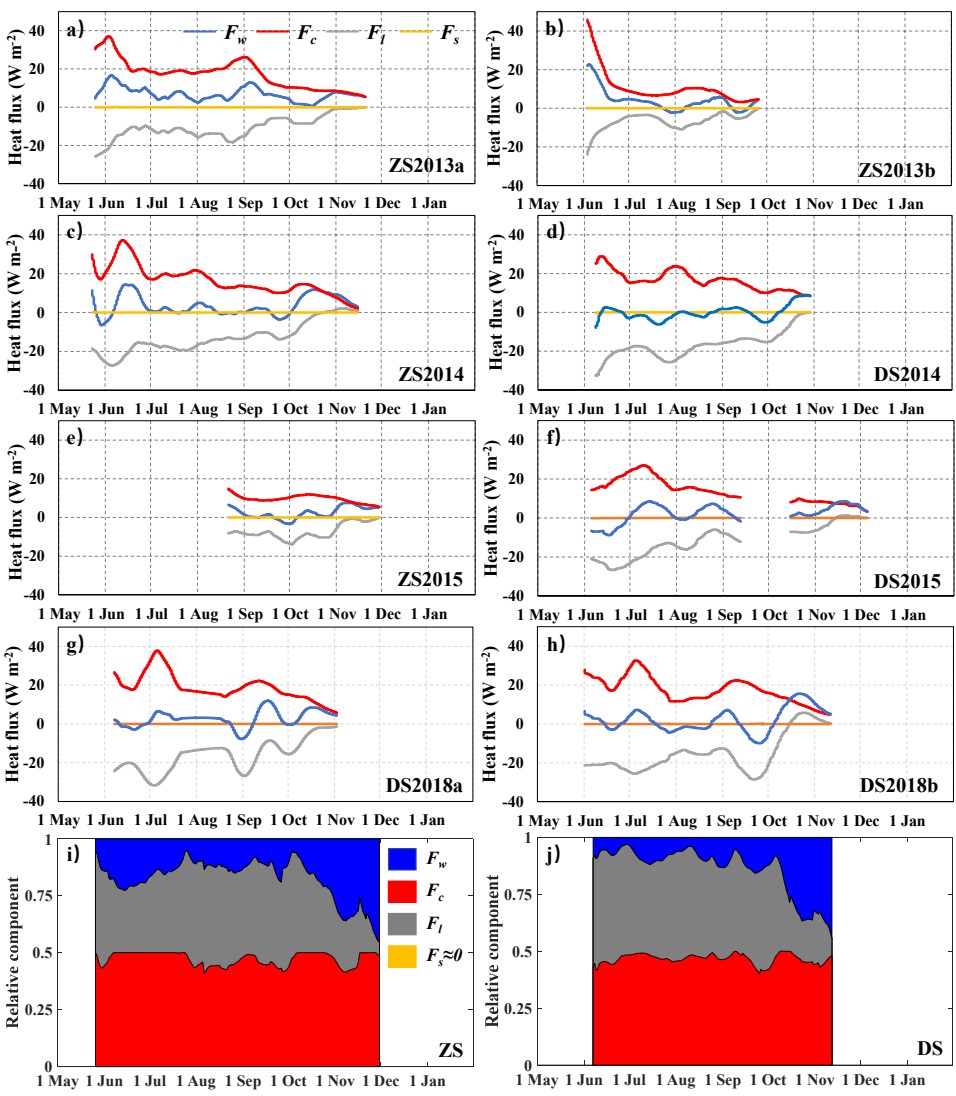

Figure 7. (a–h) Heat flux components obtained from the IMB observations at different sites, with the two panels in each row obtained in the same year, and the averaged relative contributions to the total heat balance at ice bottom of the (i) six sites at ZS and (j) five sites at DS.



## 4 Discussion

### 4.1 Comparison with early studies

The mass balance measurements of the LFI around ZS and DS given in this study (Table 1) were compared with those obtained in previous studies (Table 3); the results suggest a stable long-term condition of the LFI mass balance at both stations in the Prydz Bay. For a site close to the coast near ZS, with data from 6 years (2006 and 2012–2016), a large interannual change in the annual maximum ice thickness and the corresponding occurrence time (from early November to early December) could be identified at this site. The mean annual maximum LFI thickness was $1.62 \pm 0.14$ m (Tables 1 and 3), with the maximum

occurring in 2016 (1.82 m) and the minimum occurring in 2015 (1.49 m). The obvious discrepancy in the snow accumulation in these 2 consecutive years was the main factor contributing to the difference in the annual maximum ice thickness (Zhao et al., 2019). Similarly, at DS, combining the data used in this study (DS2016 and DS2018a) with the 26-year data from the period of 1957–2009 obtained at the same site (Heil, 2006; Heil et al., 2011), the annual maximum LFI thickness at DS did not reveal a significant trend and showed large year-to-year variations. In 2016 and 2018, the annual maximum ice thickness

was observed in mid- to late October, which was relatively early compared to the 1957–2009 average (around November 10). Our results did not follow the delayed trend obtained for the period of 1957–2009 by Heil et al. (2011). This suggests that this trend is likely not robust. At the Troll/Fimbulisen site (2005–2010; Heil et al., 2011) and the Atka Bay site (2010–2018; Arndt et al., 2020) in the eastern Weddell Sea and the McMurdo Sound "sea ice runaway" site in the Ross Sea (1986–2013; Kim et al., 2018), no significant trends were identified for either the annual maximum snow depth or the LFI thickness. This suggests

that the circum-Antarctica LFI mass balance has likely been in a relatively stable state over the past few decades. This can partly be associated with the ambiguous trends for the AT at these stations. For example, the trends of increasing AT at DS, Troll/Fimbulisen, and Neumayer (Weddell Sea) were insignificant, and the trend at McMurdo (Ross Sea) was significant only in spring with a confidence level of 0.10 (Turner et al., 2019). In addition, the role of snow accumulation complicates the response of the LFI to local climate change. Modelling work suggests that the Antarctic snow–sea ice system may be delicately

balanced, where either a decrease or an increase in the snowfall rates could lead to an increase in the ice thickness because of the regulation of the basal freezing and the snow ice or superimposed ice formation (Fichefet and Morales Maqueda, 1999; Powell et al., 2005).



Table 3. Summaries of previous studies on the landfast ice (LFI) mass balance around ZS and DS.

| Observation site | Observation method | Duration of data records | Bathymetry (m) | Maximum snow depth (m) | Maximum ice thickness (m) | Date of maximum ice thickness | References |
|---|---|---|---|---|---|---|---|
| S1~S4 | Borehole | Early Apr. to late Dec. 1992 | 11.5–200 | 0.10–0.35 | 1.60–1.74 (Ave. 1.68 ± 0.06) | Mid Nov. to mid Dec. 1992 | He et al. (1998) |
| Section 3[*1] | Hot wire thickness gauge and borehole | Late Mar. to late Dec. 2006 | 10–50 | 0.13 | 1.74 | 20 Nov. 2006 | Lei et al. (2010) |
| SIP[*1] | Borehole | 2012 to 2016 | 10 | 0.09–0.51 | 1.44–1.82 (Ave. 1.59 ± 0.15) | Early Nov. to early Dec. | Zhao et al. (2019) |
| S1[*2] | Borehole | 1957 to 2003 (20 years) | 12.4 | 0.05–0.75 | 1.44–1.98 (Ave. 1.67 ± 0.10) | Mid Oct. to late Nov. | Heil (2006) |
| S1[*2] | Borehole | 1957 to 2009 (26 years) | 12.4 | / | 1.44–1.98 (Ave. 1.70 ± 0.14) | / | Heil et al. (2011) |

Note: [*1] The locations of the site closest to the coast in Section 3 (Lei et al., 2010) and site SIP (Zhao et al., 2019) are the same as those of ZS2013a, ZS2014, and ZS2015.

[*2] The location of Site S1 (Heil, 2006; Heil et al., 2011) is the same as that of DS2016 and DS2018a.

In contrast to the unclear trend for the LFI mass balance, some statistically significant trends were found for the LFI extent and some timings of the LFI season. Compared with the maximum LFI area three decades ago, the maximum LFI area around Antarctica has decreased by approximately 10% (Fedotov et al., 1998; Li et al., 2020); this overall decreasing trend also occurred during the period from 2000 to 2018 (Fraser et al., 2021). At a more local scale, a trend toward a delayed LFI breakout date and an increased LFI duration was identified over the period of 1969–2003 in the DS region of the Prydz Bay (Heil, 2006) and over the period of 1978–2015 in the McMurdo Sound of the Ross Sea (Kim et al., 2018). Note that the LFI in different regions may respond to changes of external forcing in different ways and that the spatial scales and local geographical environment may influence the responding patterns, which may even lead to inconsistent change trends (Fraser et al., 2021). For example, the final breakout date of the Antarctic LFI is more dependent on the dynamic processes close to the shore, while the LFI extent is more dependent on the dynamic processes close to the open water. Conversely, the LFI mass balance in coastal regions depends more on local thermodynamic processes but is also affected by local small-scale dynamic processes, such as snow blowing, as shown in this study. Therefore, to thoroughly understand the complex regional patterns of the changes



in the Antarctic coastal LFI, a sustained and expanded LFI observing network, which includes monitoring of the atmospheric and oceanic boundary layers, as well as the dynamics of the adjacent cryosphere, such as changes in the iceberg distribution, the fronts of ice shelves, glaciers, and ice tongues, is essential.

**4.2 Dominant factors for the LFI thermodynamic mass balance**

In general, dynamic processes are mostly limited during the early freezing season for the Antarctic LFI. Once a solid ice cover forms, the subsequent growth of the LFI is primarily determined by thermodynamic processes that occur at the base of the ice as the heat is conducted upward and lost to the surface atmosphere. Therefore, the importance of the thermal effect of the snow depends on the thickness ratio between the snow and the ice. When removing the contributions of the snow layer by using AT_obs instead of $T_s$ for the ice thickness calculation based on the analysis model, the simulated LFI thicknesses increased

significantly at DS2014, DS2015, and DS2016 (Figs. 8g, 8h, and 8i, respectively). Such a significant increase was not found in 2018, which is attributable to the observed thin snow cover of less than 0.10 m during most of the ice season. Around ZS, when the effect of snow was not considered, the largest increase in the simulated LFI thickness, which occurred at ZS2013b, was up to 0.35 m (or 33.0%), corresponded to the highest thickness ratio of snow and ice at this site. However, even though the snow cover was relatively thick at ZS2010, the discrepancy between the simulated (without snow) and observed LFI

thicknesses in early October was relatively small at 0.24 m (or 17.4%); this was partly because the accumulation of snow mostly occurred in spring when the ice was entering the end of the growth period. During this stage with relatively thick ice, the effect of the snow became less important in reducing the ice growth rate (e.g., Merkouriadi et al., 2020).

   The sensitivity of the LFI mass balance to the AT anomaly is shown in Fig. 8. At DS, the discrepancies between the ice thickness estimated using AT_obs and that estimated using AT_mean ranged from −0.12 m at DS2018b to 0.11 m at DS2016

by the end of October because of the anomalously warm and cold winters in 2018 and 2016, respectively. The AT anomaly appears to have a larger impact on the LFI thickness at DS (4.3% on average) than that at ZS (2.0% on average). This difference can be explained by the fact that the influence of other factors, such as the oceanic heat flux and katabatic wind, on the LFI growth was relatively weak at DS compared with at ZS. During the same season, the difference between the estimated ice thicknesses using AT_obs and AT_mean was larger at ZS2013b and DS2018b with thinner ice thicknesses than at ZS2013a

and DS2018a, respectively, which indicated that the AT anomalies are more sensitive with thin ice because of its weak thermal inertia (Lei et al., 2018). To assess the influence of climate change on the mass balance of Antarctic LFI, the seasonality of climate change should be considered because of the large seasonal differences in the response of sea ice growth to climate change.

   Comparing the calculation results when considering or ignoring the oceanic heat flux in the analysis model, we found that

oceanic forcing appears to exert a larger influence on the mass balance of the LFI near ZS than near DS, as shown in Figs. 7i and 7j. Limited by the shallow water depth around DS, it is almost impossible for deep warm water to intrude into the coastal region, which leads to small values and contributions of the oceanic heat flux to the LFI mass balance there. The averaged



oceanic heat flux at DS was estimated to be 1.6 W m$^{-2}$ during the period of June–October for the years from 2014 to 2018, which was comparable to the value (1.4 W m$^{-2}$) obtained from 1980 to 1985 (Heil et al., 1996). After removing the oceanic

heat flux component in the analysis model, the simulated ice thicknesses at the buoy sites near DS increased by 5.7% on average by the end of October. Correspondingly, at ZS, the simulated ice thicknesses increased at most of the buoy sites when parameterizing $F_w$ to zero, with the largest increase being 0.46 m (or 34.1%) obtained at ZS2013a (Fig. 8b). However, ZS2009 and ZS2010 were two exceptions, with the simulated ice thicknesses at both sites decreasing by 0.07 m after removing the contributions of the oceanic heat flux (Figs. 8a and 8c). Located within an area of grounded icebergs that have calved from

Dålk Glacier (Fig. 1b), the seawater at a depth of 2 m at both sites was colder than the freezing point throughout the LFI growth period from July to the end of October, with an average temperature of −2.0 °C. One possible explanation for the exceptions at ZS2009 and ZS2010 is that the seawater at these sites was likely supercooled and potentially related to the outflow of glacial meltwater, which resulted in a negative oceanic heat flux that was conducive to the LFI growth. Such a mechanism has been observed in LFI in front of some Antarctic ice shelves (e.g., Langhorne et al., 2015). To further understand the influence

mechanism of supercooled glacial meltwater on the LFI mass balance, it is recommended to carry out simultaneous observations of the sea ice mass balance and the oceanography of the ocean stratification, heat content, and currents at the glacial front. Isotopic analyses of the under-ice seawater are also helpful to identify the influence range of glacial meltwater in the LFI zone.


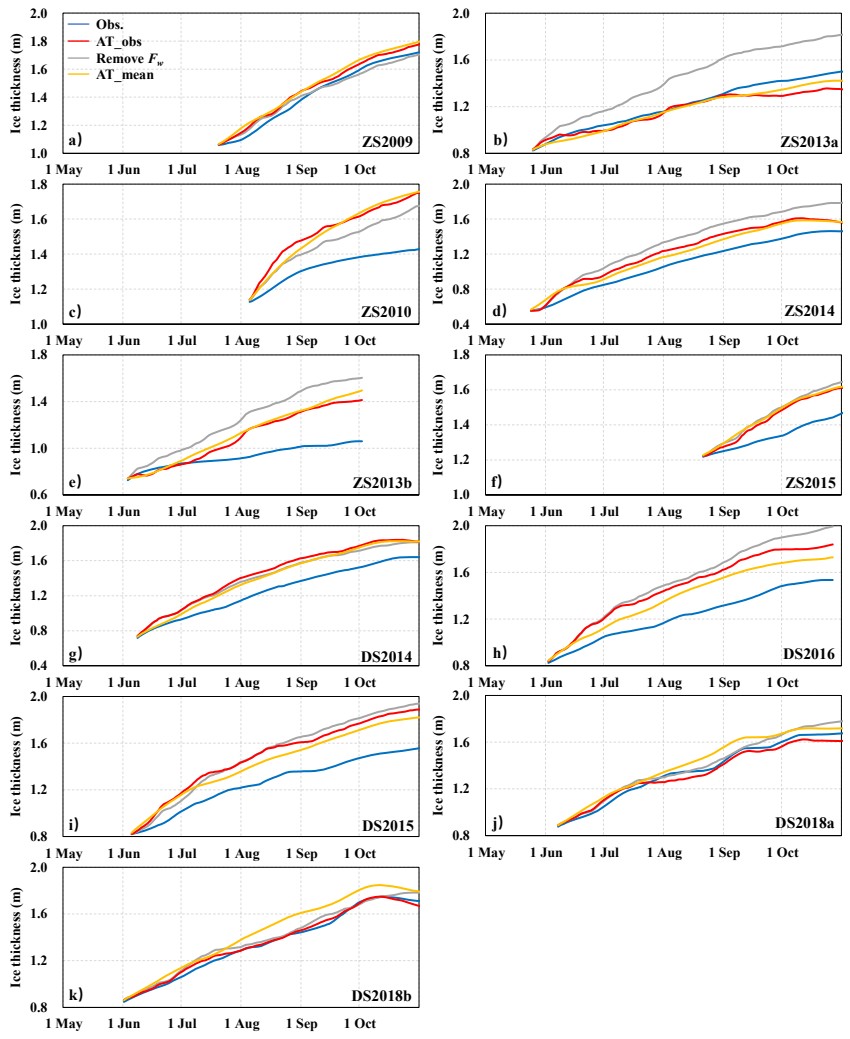

Figure 8. LFI thicknesses observed by the buoys (blue line) and calculated ignoring the oceanic heat flux (gray line), using AT_obs (red line) and AT_mean (yellow line).



### 4.3 Flooding and snow ice formation

When the snow mass is sufficiently large, the ice surface submerges and slush is formed. Snow ice formation depends on the evolution of the ice freeboard and ice permeability (Eicken et al., 1994, 1995). At our study sites, the estimated freeboards decreased as the snow accumulated and negative freeboards were identified at ZS2013b (Fig. 9a) and ZS2010 (Fig. 9c), suggesting the possibility of flooding. Ice surface flooding is controlled by upward brine percolation when the ice layer is warmer than a critical temperature (Golden et al., 1998) or through macroscopic fractures when the ice surface is depressed

because of local ice deformation (Lei et al., 2022). Seasonally, ice bulk temperatures usually reach the critical temperature of −5 °C around early November. At ZS2013b, a snow storm occurred in early July and its resultant loading pushed the snow–ice interface below sea level (Fig. 9a). The impermeable layer, defined by temperatures below −5.0 °C, was warmed at the onset of the storm and the LFI layer became permeable. Flooding occurred and the slushy layer froze during the subsequent cold period forming snow ice. The obvious flooding event occurred in late August and lasted for approximately half a month.

We can clearly see that the snow–ice interface shifted upward approximately 0.20 m based on the HT ratio profile obtained by the SIMBA thermistor chain measurements (Fig. 2b). Until the end of the observation in early October, the contribution of snow ice to the LFI mass balance reached 0.40 m at ZS2013b, approximately 27% of the total ice thickness (the thickness of snow ice included). This proportion of snow ice was comparable to the results from previous studies of the LFI around ZS. On the basis of the ice core samples collected from the Nella Fjord, Tang et al. (2007) found that the regional average proportion

of snow ice varied from 8% to 38% by the end of the ice growth season in late January 2003. Using a sea ice thermodynamic model, Zhao et al. (2019) estimated that snow ice contributed to 4–23% of the total maximum ice thickness for the LFI off ZS in 2012. Similar to ZS2013b, snow ice formation was very likely to occur at ZS2010 after mid-November, during a period when the snow–ice interface was approximately 6 cm below the sea level and the ice layer became permeable (Fig. 9d). The formation of snow ice was simulated by Zhao et al. (2020) for LFI around ZS in 2015. However, neither the ice temperature

nor the HT ratio profile at ZS2015 in our study shows signs of snow ice formation, which is likely because drifting snow was not considered in the sea ice model used by Zhao et al. (2020) and the simulated snow depth was much larger than the observed value at ZS2015.

Our buoy data did not indicate any signs of the formation of snow ice over the LFI around DS, consistent with the results presented by Heil (2006). This was likely related to the relatively thin snow cover there. Therefore, the flooding and formation

of snow ice in the LFI zone was not widespread, which differs from the drifting ice zone over the Southern Ocean (e.g., Eicken et al., 1994; Jeffries et al., 2001; Maksym and Markus, 2008). The dynamic deformation of LFI occurs only over a very small spatial range, together with the relatively small thickness ratio between the snow and the LFI compared with the drifting ice zone (e.g., Ozsoy-Cicek et al., 2013), which are likely the main reasons explaining why flooding and snow ice formation are not as widespread over Antarctic LFI as in the drifting ice zone.


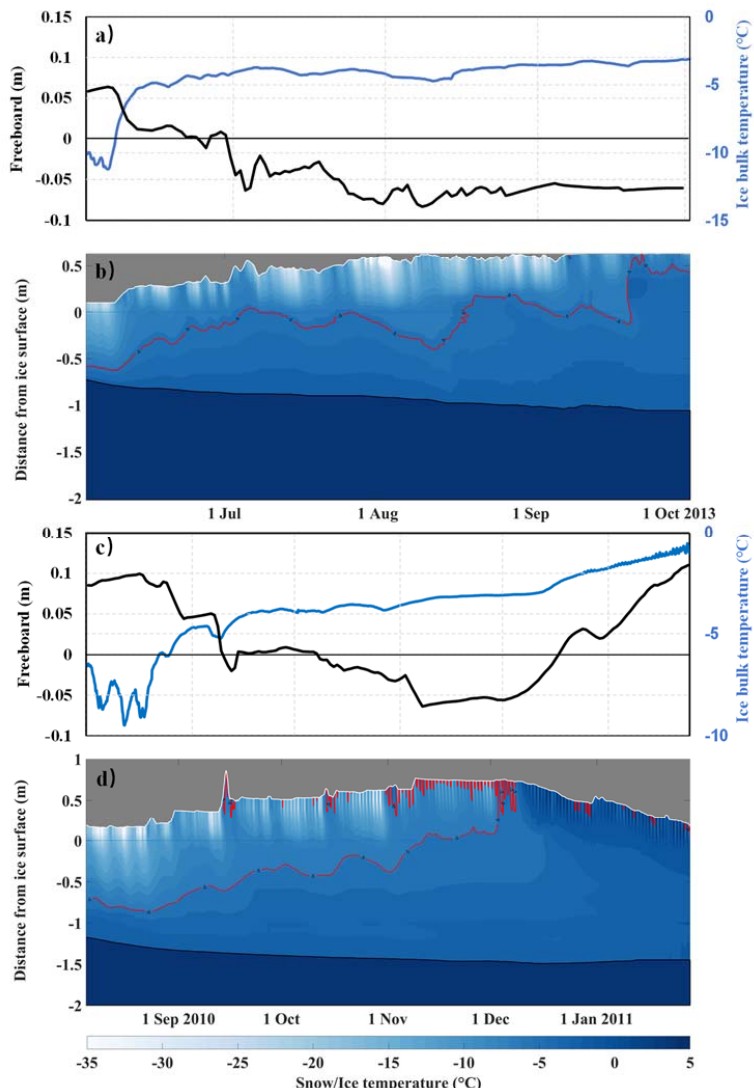


Figure 9. Estimated sea ice freeboard (black line, left axis), ice bulk temperature (blue line, right axis), and temperature contours in the snow and sea ice measured by (a) and (b) ZS2013b and (c) and (d) ZS2010. The white, red, and black lines in panels (b) and (d) represent the snow surface, the −5 °C isotherm, and the ice base, respectively.



During spring and summer, Antarctic sea ice melts primarily from the bottom up because of the relatively large oceanic heat flux, as well as the cool and windy surface conditions, which prevent melt-pond formation (Nicolaus et al., 2009; Arndt et al., 2021). Compared with the snow ice, the formation of superimposed ice was less extensive because of the lack of a supply of meltwater. With the extensive surface thaw–refreeze cycles after mid-November in 2013, the upward shift of the snow–ice interface at ZS2013a provided evidence of the possibility of the formation of superimposed ice. At this site, the snow–ice

interface was consistently above the water level, the snow meltwater percolated downward to the colder snow–ice interface where the superimposed ice formed. The superimposed ice formed in early November and lasted until 27 November at ZS2013a with a maximum thickness reaching 0.10 m (Fig. 2a). The contribution of the superimposed ice to the LFI mass balance in our study area can be considered to be weaker than that of the snow ice. However, the formation of superimposed ice can prevent the further downward infiltration of snow meltwater, which is expected to refreeze into ice plugs within the

pores of the sea ice (Polashenski et al., 2012). Therefore, superimposed ice over the parental ice layer, while having a short life and a small thickness, would provide nontrivial influences on the LFI thermodynamic processes.

## 5 Conclusions and outlook

Regional, seasonal, and interannual variations in the LFI mass balance in the Prydz Bay, East Antarctica, were investigated using IMB data observed over 7 ice seasons during the periods of 2009–2010, 2013–2016, and 2018. The observations were

scattered in two domains (ZS and DS) oriented along a west–east distribution with a separation of 110 km and ranging between 100 m and 6 km from the shore. A larger local spatial variability of the snow depth and ice thickness was observed over the LFI zone at ZS than at DS. The LFI around DS grew faster and reached a larger maximum ice thickness of 1.64 ± 0.08 m by mid-October to mid-November, which was approximately 20 days earlier than the occurrence of the maximum ice thickness around ZS (1.59 ± 0.17 m). We found significant interannual variability in the LFI mass balance at both the ZS and DS sites.

This coincides with the LFI in the Weddell and Ross seas, where no clear steady trend of the LFI mass balance has been identified (Heil et al., 2011; Kim et al., 2018; Ardnt et al., 2021).

The snow accumulation showed a distinct pattern between the ZS and DS domains. During the observation period, the maximum snow depth around ZS and DS had ranges of 0.11–0.76 and 0.11–0.38 m, respectively. This difference was partly due to local differences in the solid precipitation, as well as the distribution and morphology of the grounded icebergs

associated with the wind regime, which played a critical role in the distribution of the snow accumulation further affecting the growth of the LFI. Formation of snow ice was observed at ZS2013b, a site offshore of ZS, contributing up to 27% of the total ice thickness. However, snow ice did not prevail in the LFI region of the Prydz Bay, particularly close to the shore, where snow was drifted away by strong continental wind. This is obviously different from the drifting ice zone in the Southern Ocean



where extensive snow ice was found because of the relatively small snow–ice thickness ratio and strong sea ice deformation
(e.g., Ozsoy-Cicek et al., 2013).

The AT anomaly during the observed period had a more profound impact on the LFI growth at DS than at ZS. We believe this is because the oceanic heat flux was smaller at DS than at ZS, especially during the early ice rapid growth stage, which is likely related to the difference in the nearshore terrestrial morphology and coastal bathymetry at the two locations. The supercooled glacier meltwater supplied from the Dålk Glacier likely reduced the local oceanic heat flux at specific sites in the
ZS domain further enhancing the spatial and interannual variability of the LFI mass balance. Our residual analyses suggest that the localized seasonal variation of the oceanic heat flux partly regulated the seasonality of the LFI mass balance; this was also supported by modelling results (e.g., Yang et al., 2015). However, there are still some oceanographic processes, such as the ocean tides, vertical mixing, supercooling, brine rejection, and/or outflow of glacier meltwater, that remain to be further quantified to estimate the turbulent heat exchange between the LFI and the ocean.

The heterogeneity of LFI in Antarctica is relatively inconspicuous compared with that in the drifting ice zone in the Southern Ocean. Nevertheless, we believe numerous small-scale factors such as the snow dynamics, surface morphology, and coastal bathymetry, as well as the local distributions of the ice shelves, glaciers, ice tongues, and grounded icebergs, contribute significantly to the seasonal and interannual variability of the LFI mass balance. These factors should therefore be better parametrized in regional climate models. This will allow for better representations of the LFI boundary conditions using high-
resolution models (Liu et al., 2022).

In Antarctica, LFI observations rely on research base operations and are often interrupted largely as a result of the harsh environment and safety regulations. IMBs are currently the only automated instrument that can yield the LFI mass balance at a seasonal scale. Nevertheless, challenges facing intricately detailed measurements of the LFI mass balance, including internal gap layers that can only be detected manually (Zhao et al., 2022), still remain. To capture the coupling process between the
sea ice mass balance and ice-associated ecosystems, better integrated buoys are required to make synchronous measurements of the sea ice mass balance, biogeochemical, and optical parameters, for example, the multi-sensor Unmanned Ice Station (Lei et al., 2022). Further developments of the LFI monitoring instrumentation and observation network (AFIN) are urgently needed to better understand the various physical–biochemical processes and the importance of LFI to the Antarctic system, especially in coastal regions.

**Data Availability**

All presented IMB data have been submitted and will be published in PANGAEA at the time of paper publication. The hourly meteorological parameters at ZS and DS and the daily solid precipitation at the Russian Progress II station were obtained from the global Integrated Surface Hourly (ISH) database, https://www.ncei.noaa.gov/data/global-hourly/ (last access: 30 June 2022).





**Author Contributions**

NL and RL developed the concepts and the approach and gathered and prepossessed the buoy data. NL, RL, PH, BC, and BL performed data extraction and analysis. All co-authors participated in the writing and/or revision and approval of the submitted manuscript.

**Competing interests**

Two of the co-authors are members of the editorial board of *the Cryosphere*, and the authors declare that they have no other conflict of interest.

**Acknowledgements**

This work was carried out and the data used in this manuscript were produced as part of the Chinese National Antarctic Research Expedition (CHINARE) and Australian National Antarctic Research Expeditions (ANARE). We are very grateful to
the overwintering teams at ZS and DS from 2009 to 2018 for the measurements they conducted on the landfast ice.

**Financial support**

This work was supported by the National Key Research and Development Program of China (2018YFA0605903) and the National Natural Science Foundation of China (52192691 and 41606222). PH was supported by the Australian Government through Australian Antarctic Science project 4506 and the International Space Science Institute Grant 406. BC was
supported by the European Commission H2020 Project Polar Regions in the Earth System (PolarRES, Grant 101003590).

**Review statement**

This paper was edited by XX and reviewed by XX anonymous referees.

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
