# Peer review of "Seasonal and interannual variability of the landfast ice mass balance between 2009 and 2018 in Prydz Bay, East Antarctica"

_The Cryosphere, 2022_

## Referee Comment (RC2)

Review for tc-2022-198

Landfast ice (LFI) is one of the predominant features around the Antarctic coastal zone, representing 4-12.8% of the Antarctic sea ice extent while around 28% of the total sea ice volume in the Antarctica. LFI is a sensitive indicator to climate change. Its local and region variations are affected by the atmospheric and oceanic conditions, as well as the local conditions. Prydz Bay in East Antarctica is the third largest bay around the Antarctic continent, with the Chinese Zhongshan Station and the Australian Davis Station in this bay. With the IMB data near the two stations between 2009 and 2018, Li et al. studied the "Seasonal and interannual variations in the landfast ice mass balance between 2009 and 2018 in Prydz Bay, East Antarctica". They presented the LFI differences between the two stations, identified the differences are due to local differences in topography and katabatic wind regime, and investigated the main factors regulating the LFI mass balance. The manuscript is well structured, and generally well written. More detail comments are given below. After the minor revision, I recommend to publish this manuscript.

Detail comments:

L67, "Of these" can be remove.

L69, Suggest move "the third largest bay around the Antarctic continent" after L67 within the Prydz Bay.

L72-73, "Largely as a result of discontinuous observations associated with logistic difficulties" can be rewritten for concise, e.g., "Due to logistic difficulties for regular observations"

L74-75, Is it "the deployment of the ice mass balance buoy" or "the ice mass balance buoy" permit the continuous monitoring of the sea ice mass balance. Suggest rewrite to "The ice mass balance buoys (IMBs) permit the continuous monitoring of the sea ice mass balance and their deployments are human resources economy.

When the IMBs started to deploy? Please add this information.

L85, Fig. 1: Enlarge the red stars symbols in Fig. 1a.
        Can you rotate b) and c) 180º to let stations on the upper right and the sea ice or sea ice/ ocean on the lower left?
        There are two red stars in Fig. 1b. Are they same as in Fig. 1a?

L106, remove "observed" or "record"

L115, Table 2, Add a column for the type of IMB (CRREL-IMB or SIMBA) deployed

L133, Label "the Russian Progress II station" on Fig. 1.

L223, From Fig. 3d, the wind are dominated by easterly wind and ESE wind at ZS, and dominated by NE, and NNE and ENE at DS station.

L231, where is the Vestfold Hills? Can you label it in Fig. 1?

L249, remove "obtained" and replace "synchronously" with "synchronous".

L265, Fig. 4, No lines for ZS2014 in Fig. 4a.

L318, replace "; the values" with "which"

L315, Fig. 6b, is it difficult to see the temperature change in S1 and S2, please change another color for them. Or add one figure for temperature gradient? This is related to your statement in L348-349.

L358, "This temporal lag …. within the ice column". Give a time for this temporal lag change".

L365, I disagree with "the basal ice growth was primarily regulated by Fc and Fw". From Fig.7, the basal ice growth was also regulated by Fl. So please also discuss the Fl in the manuscript.

L370, replace ";, this drove" with "which drove"

L402, "For a site", point out this site is which site in Table 3

L406, the same site, I think you are pointing to S1 in Heil (2006) and Heil et al. (2011). Please add this information in your text.

L429-443, Your LFI mass balance results are from the IMBs point measurements. The point measurements are more related to small-scale processes. How are you related the small-scale results to local-scale, regional-scale? Can you discuss this a little bit?

L448-451, Obviously your description is around DS. Could you add this information clearly in the text?

L452, the largest increase in the simulated LFI thickness, as I see from Fig. 8, occurred at ZS2010 (Fig. 8c), not ZS2013b (Fig. 8e). Can you re-check your results?

L461-462, This sentence can be rewritten as "This difference indicates that the LFI at ZS was more influenced by other factors, such as the oceanic heat flux and katabatic wind, compared at DS.

L470-471, Please make sure that you refer to the right Figures, Fig. 7 or Fig. 8? In Fig. 7i and 7j, one can see the larger influence of FW near ZS than near DS. But you are comparing with and without the oceanic heat flux, Fig 8 might be the right figure you refer to.

L472, Make the sentence to concise. Such as rewrite as "which leads to small contributions to the oceanic heat flux to the LFI mass balance there."

L475-479, It seems that your increase or decrease of simulated thickness is compared to AT_obs or AT_mean. Please clarify this in your text.

L490, using same y-axis ticks in all the subplots for Fig. 8.

L523, remove "explaining" before "why"

L533, "upward shift" or "downward shift", please recheck. From Fig.2a, the snow-ice interface was downward shift.

L555, not only distribution of snow but also redistribution of snow. Please add this information in your text also.

---

## Author Comment (AC1)

**Response to RC1**

Thank you for your time and constructive comments on the manuscript "Seasonal and interannual variations in the landfast ice mass balance between 2009 and 2018 in Prydz Bay, East Antarctica". We would consider each comment carefully and incorporate practically all of them in the revised manuscript.

**Major comments**

What does the negative $F_w$ mean as shown in Figure 7? In general, the ice base temperature is higher than the sea water temperature, which indicates the positive $F_w$. Does the significantly negative $F_w$ occurred in DS2015 and DS2018b mean the existence of supercooled water? Or is it just a modelled error? How large are the modelled errors for heat flux components? If it is difficult to quantify these errors, the uncertainties of modelling results should be discussed at least.

Reply: The oceanic heat flux in Figure 7 is derived from the heat flux residual method, and its minimum, sometimes negative, usually occurred in late September or early October near ZS (Lei et al., 2010; Yang et al., 2015). Given that the nearest glacier, Søsdal Glacier, is about 12 km south of DS and in the absence of simultaneous oceanic measurements, we cannot ascertain that the significantly negative $F_w$ in DS2015 and DS2018b originates from supercooled water. These small negative $F_w$ can also be partly attributable to the potential estimation uncertainty (1-2 W m$^{-2}$, Lei et al., 2014). To address this still open issue, we recommend to combine the observation of under-ice turbulence in the future to improve estimation accuracy of ocean heat flux and clarify whether there will be a negative value really associated with supercooled water. In the revised manuscript, we will add some relative discussions.

Ref.: Lei, R., N. Li, P. Heil, B. Cheng, Z. Zhang, and B. Sun (2014), Multiyear sea-ice thermal regimes and oceanic heat flux derived from an ice mass balance buoy in the Arctic Ocean, J. Geophys. Res. Oceans, 119, doi:10.1002/2012JC008731.

I realize that this study provides abundant helpful information about LFI evolution based on observations. However, these findings are not well summarized. I would suggest the authors to add a sketch map to summarize the key findings and related mechanism, especially for describing the critical factors/ thermodynamic processes that are responsible for the LFI variabilities.

Reply: Good suggestion. A sketch map is very helpful to summarize the findings. We will add a sketch map in section 5 to compare the characteristics of LFI near ZS and DS, and also point out the critical factors and thermodynamics processes that are responsible for the LFI variations.

**Specific comments:**

Figure1: the expression in Figure 1(a) could be easily misunderstood. The whole Antarctica and the study region in east Antarctica should be given separately.

Reply: To avoid misunderstanding, the map of the Antarctica and the study region in Figure 1a will be separated and given as two panels.

Table2: add a column to present the type of buoys.
Reply: We think you are referring to Table 1. We will add the type of IMBs to this table as your suggestion.

An additional table is needed to summarize the observed variables of each buoy, and give the corresponding key technical specifications (e.g., precision, uncertainty, measurement range).
Reply: The technical details concerning CRREL-IMB and SIMBA as well as the key technical specifications of the observed variables for these two types of IMBs can be found in Richter-Menge et al. (2006) and Jackson et al. (2013). For easy reference, we will add a table S1 in the supplement to summarize the above-mentioned information.

What does vertical red bars represent in Figure 6b and Figure 9d?
Reply: The red lines in Figure 6b and Figure 9d represent the isotherm at −5 °C, which is defined as the threshold temperature of the potential percolation phase transition (Golden, 1998). As the diurnal cycle in air temperature becomes more outstanding since late September, which could further affect the upper sea ice, the −5 °C isotherms start to cluster together and look like red bars. To avoid misunderstanding, we will optimize these two illustrations in the revised manuscript.

---

## Author Comment (AC2)

**Response to RC2**

Thanks for your time and constructive comments on the manuscript "Seasonal and interannual variations in the landfast ice mass balance between 2009 and 2018 in Prydz Bay, East Antarctica". We will consider each comment carefully and incorporate practically all of them.

**Specific comments:**

L74-75, Is it "the deployment of the ice mass balance buoy" or "the ice mass balance buoy" permit the continuous monitoring of the sea ice mass balance. Suggest rewrite to "The ice mass balance buoys (IMBs) permit the continuous monitoring of the sea ice mass balance and their deployments are human resources economy.
When the IMBs started to deploy? Please add this information.
Reply: We will rewrite this sentence and make it clearer. Actually, we want to say "The ice mass balance buoys (IMBs) permit the continuous monitoring of the sea ice mass balance and their deployments since the end of the 1990s in both Arctic and Antarctic provide a crucial tool for monitoring sea ice changes".

L85, Fig. 1: Enlarge the red stars symbols in Fig. 1a.
Can you rotate b) and c) 180º to let stations on the upper right and the sea ice or sea ice/ocean on the lower left?
There are two red stars in Fig. 1b. Are they same as in Fig. 1a?
L133, Label "the Russian Progress II station" on Fig. 1.
L231, where is the Vestfold Hills? Can you label it in Fig. 1?
Reply: Thank you for your comments. The red stars in Fig. 1a indicate the Chinese Zhongshan Station and the Australian Davis Station, while the red stars in Fig. 1b indicate the Chinese Zhongshan Station and the Russian Progress II Station. They are not the same. To make Fig. 1 clearer, we will modify this illustration, including to separate Fig. 1a into two pannels, enlarge the red star symbols in Fig. 1a, rotate Fig. 1b and Fig. 1c as suggestion, label "the Russian Progress II Station" and "Vestfold Hills" etc.

L115, Table 2, Add a column for the type of IMB (CRREL-IMB or SIMBA) deployed
Reply: We think you are referring to Table 1. We will add a column for the type of IMB to this table following your suggestion.

L223, From Fig. 3d, the wind are dominated by easterly wind and ESE wind at ZS, and dominated by NE, and NNE and ENE at DS station.
Reply: Thank you for pointing this out. The wind forcing at ZS is characterized by katabatic winds, of which, winds from the east, ENE and ESE are dominant, with the frequency of 38.9%, 22.8%, and 11.7%. The wind forcing at DS is largely driven by passing synoptic systems, with the dominant wind direction from NNE to ESE, accounting 66%. In addition to the distribution of wind speed, we will also add the distribution of wind direction to Fig. 3d.

L265, Fig. 4, No lines for ZS2014 in Fig. 4a.

Reply: The topmost temperature thermistor of ZS2014 was just placed on the snow–ice interface at deployment as a result of an inaccurate operation. The snow depth of ZS2014 could not be retrieved from the temperature profiles and the in situ measurements were used instead. Only 9 observations of snow depth were made during the operation period of ZS2014. In Fig.4a, these snow depth measurements are shown as blue hollow dots, but not a line. We will add this information in the caption of Fig. 4.

L315, Fig. 6b, is it difficult to see the temperature change in S1 and S2, please change another color for them. Or add one figure for temperature gradient? This is related to your statement in L348-349.

Reply: We will add a subplot of temperature gradient as Figure 6c so that the temperature change can be clearly identified during the typical winter warming events, S1 and S2.

L358, "This temporal lag …. within the ice column". Give a time for this temporal lag change".

Reply: During S1 (mid July), this temporal lag was about 3 days and it reached 7 days by late August and early September (S2). We will specify this temporal lag change in Section 3.4.

L365, I disagree with "the basal ice growth was primarily regulated by $F_c$ and $F_w$". From Fig.7, the basal ice growth was also regulated by $F_l$. So please also discuss the Fl in the manuscript.

Reply: In the formula of heat balance at the ice base, $F_l$ is defined as the equivalent latent heat flux and calculated based on the ice growth/melt rate. Thus, the $F_l$ is actually directly determined by the sea ice growth rate, rather than playing a regulation role. We will make the expression clearer.

L402, "For a site", point out this site is which site in Table 3
L406, the same site, I think you are pointing to S1 in Heil (2006) and Heil et al. (2011). Please add this information in your text.

Reply: "For a site close to the coast near ZS", this site refers to the site closest to the coast in Section 3 and the site SIP in Table 3, and also the same site of ZS2013a, ZS2014 and ZS2015. "Similarly, at DS…from the period of 1957-2009 obtained at the same site", the same site here refers to site S1 in Heil (2006) and Heil et al. (2011). To make the expression clearer, we will add these information in Table 3 and in the Section 4.1 as suggested.

L429-443, Your LFI mass balance results are from the IMBs point measurements. The point measurements are more related to small-scale processes. How are you related the small-scale results to local-scale, regional-scale? Can you discuss this a little bit?

Reply: We will add some discussions on the representativeness on our measurements and how to upscale the derived results.

L448-451, Obviously your description is around DS. Could you add this information clearly in the text?
Reply: This part does describe the effect of snow layer on the LFI thickness near DS. We will specify this location information in the Section 4.2.

L452, the largest increase in the simulated LFI thickness, as I see from Fig. 8, occurred at ZS2010(Fig. 8c), not ZS2013b (Fig. 8e). Can you re-check your results?
L490, using same y-axis ticks in all the subplots for Fig. 8.
Reply: In Fig. 8, the y-axis ticks are different, which would be misleading that the largest increase in the simulated LFI thickness occurred at ZS2010. In fact, when the effect of snow was not considered, the largest increase in the simulated LFI thickness at ZS2010 was 0.23m, less than that at ZS2013b (0.35m). To avoid this misunderstanding, we will use the same y-axis ticks in Fig. 8 as suggested.

L470-471, Please make sure that you refer to the right Figures, Fig. 7 or Fig. 8? In Fig. 7i and 7j, one can see the larger influence of $F_w$ near ZS than near DS. But you are comparing with and without the oceanic heat flux, Fig 8 might be the right figure you refer to.
Reply:  We want to illustrate the oceanic heat flux exerts a larger influence on the mass balance of the LFI near ZS than near DS, which could also be seen in Figs. 7i and 7j. We will rewrite this sentence to make the expression clearer.

L472, Make the sentence to concise. Such as rewrite as "which leads to small contributions to the oceanic heat flux to the LFI mass balance there."
Reply: To make the expression clearer, we will rewrite this sentence as "which leads to small oceanic heat flux and its small contribution to the LFI mass balance there."

L475-479, It seems that your increase or decrease of simulated thickness is compared to AT_obs or AT_mean. Please clarify this in your text.
Reply: To assess the effect of the oceanic heat flux on the LFI growth, we compared the evolution of the ice thickness estimated by taking into account the oceanic heat flux using Eq. (4) to those estimated ignoring this flux. To identify the impact of snow cover on the LFI mass balance from the perspective of the thermal insulation effect, we used the AT obtained from the year of observation (AT_obs) instead of Ts for the LFI thickness calculation. The forcing using AT_obs actually ignores the attenuation effect of snow cover on air temperature. We will make the expression clearer.

L533, "upward shift" or "downward shift", please recheck. From Fig.2a, the snow-ice interface was downward shift.
Reply: It is a mistake. We will correct it.

L555, not only distribution of snow but also redistribution of snow. Please add this information in your text also.

Reply: Good point. We will add the mechanism of snow redistribution to explain our results in the revied manuscript.

L67, "Of these" can be remove.

L69, Suggest move "the third largest bay around the Antarctic continent" after L67 within the Prydz Bay.

L72-73, "Largely as a result of discontinuous observations associated with logistic difficulties" can be rewritten for concise, e.g., "Due to logistic difficulties for regular observations"

L106, remove "observed" or "record"

L249, remove "obtained" and replace "synchronously" with "synchronous".

L318, replace "; the values" with "which"

L370, replace ";, this drove" with "which drove"

L461-462, This sentence can be rewritten as "This difference indicates that the LFI at ZS was more influenced by other factors, such as the oceanic heat flux and katabatic wind, compared at DS.

L523, remove "explaining" before "why"

Reply: All the grammatical mistakes and inappropriate expressions will be revised as suggestions.

---

## Author Response (AR1)

**Responses to editor**

Dear Dr. Homa Kheyrollah Pour,

Thank you for handling our manuscript. We have made a thorough revision of our manuscript. Please see below a short summary on our revision:

1 adding a sketch map to summarize the key findings and related mechanism;
In the section of conclusions and outlook, we added a sketch map to better summarize and compare the seasonal and interannual characteristics of Landfast ice (LFI) between stations in Prydz Bay, with the critical factors that are responsible for the LFI variations. (Figure 10 and Lines 565-568, 570-572)

2 adding the DOI number of the buoy data;
The IMB data have been published in PANGAEA. We added all the DOI numbers in the section of data availability. (Lines 612-615)

3 resolving contradictions noted by the reviewers;
Resolved accordingly, for details, see the reply to the reviewer's comments.

4 revising the figures according to specific comments of the reviewers.
Revised accordingly, for details, see the reply to the reviewer's comments.

Please find below point by point our responses (black text) to the comments (blue text) from reviewers.

Thank you for your time,
Best regards,
Ruibo Lei and co-authors

Thank you for your time and constructive comments on the manuscript "Seasonal and interannual variations in the landfast ice mass balance between 2009 and 2018 in Prydz Bay, East Antarctica". We have considered the comments carefully and modified our manuscript accordingly as part of this revision.

**Major comments**

What does the negative $F_w$ mean as shown in Figure 7? In general, the ice base temperature is higher than the sea water temperature, which indicates the positive $F_w$. Does the significantly negative $F_w$ occurred in DS2015 and DS2018b mean the existence of supercooled water? Or is it just a modelled error? How large are the modelled errors for heat flux components? If it is difficult to quantify these errors, the uncertainties of modelling results should be discussed at least.

The oceanic heat flux in Figure 7 is derived from the heat flux residual method, and its minimum, sometimes negative, usually occurred in late September or early October near ZS (Lei et al., 2010; Yang et al., 2015). Given that the nearest glacier, Søsdal Glacier, is about 12 km south of DS and in the absence of simultaneous oceanic measurements[#], we cannot ascertain that the significantly negative $F_w$ in DS2015 and DS2018b originates from supercooled water. These small negative $F_w$ can also be partly attributable to the potential estimation uncertainty (1-2 W m$^{-2}$, Lei et al., 2014) related to improper parameterizations using in the estimation. To address this still open issue, we recommend to combine the observation of under-ice turbulence and detail measurement of sea ice physical parameters in the future to improve estimation accuracy of ocean heat flux and clarify whether there will be a negative value really associated with supercooled water. In the revised manuscript, we added some relative discussions in the Section 3.4. (Lines 370-372, 395-401).

I realize that this study provides abundant helpful information about LFI evolution based on observations. However, these findings are not well summarized. I would suggest the authors to add a sketch map to summarize the key findings and related mechanism, especially for describing the critical factors/ thermodynamic processes that are responsible for the LFI variabilities.

We added a schematic diagram in section 5 to compare the characteristics of LFI near ZS and DS, and also point out the critical factors and thermodynamics processes that are responsible for the LFI variations (Figure 10 and Lines 565-567, 571-573).

**Specific comments:**

Figure1: the expression in Figure 1(a) could be easily misunderstood. The whole Antarctica and the study region in east Antarctica should be given separately.

We modified this illustration according to the suggestion. The map of the Antarctica and the study region in Figure 1a was separated and given as two panels. (Figure 1 and Lines 93-97)

Table2: add a column to present the type of buoys.

We think you are referring to Table 1. We added the type of IMBs to Table 1 according to the suggestion. (Table 1)

An additional table is needed to summarize the observed variables of each buoy, and give the corresponding key technical specifications (e.g., precision, uncertainty, measurement range).

The technical details concerning CRREL-IMB and SIMBA as well as the key technical specifications of the observed variables for these two types of IMBs can be found in Richter-Menge et al. (2006) and Jackson et al. (2013). For easy reference, we added a table S1 in the supplement to summarize the above-mentioned information (Table S1).

What does vertical red bars represent in Figure 6b and Figure 9d?

The red lines in Figure 6b and Figure 9d represent the isotherm at −5 °C, which is defined as the threshold temperature of the potential percolation phase transition of sea ice (Golden, 1998). As the diurnal cycle in air temperature becomes more outstanding since late September, which could further affect the upper sea ice, the −5 °C isotherms start to cluster together and look like red bars. We've clarified this in the captions of Figure 6 and Figure 9.

Thanks for your time and constructive comments on the manuscript "Seasonal and interannual variations in the landfast ice mass balance between 2009 and 2018 in Prydz Bay, East Antarctica". We have considered the comments carefully and modified our manuscript accordingly as part of this revision.

**Specific comments:**

L67, "Of these" can be remove.

L69, Suggest move "the third largest bay around the Antarctic continent" after L67 within the Prydz Bay.

L72-73, "Largely as a result of discontinuous observations associated with logistic difficulties" can be rewritten for concise, e.g., "Due to logistic difficulties for regular observations"

We corrected these grammatical mistakes and inappropriate expressions according to the comments. (Lines 68; 69-70; 73-74)

L74-75, Is it "the deployment of the ice mass balance buoy" or "the ice mass balance buoy" permit the continuous monitoring of the sea ice mass balance. Suggest rewrite to "The ice mass balance buoys (IMBs) permit the continuous monitoring of the sea ice mass balance and their deployments are human resources economy.

When the IMBs started to deploy? Please add this information.

We rewrote this sentence according to the suggestions. (Lines 76-78)

L85, Fig. 1: Enlarge the red stars symbols in Fig. 1a.

Can you rotate b) and c) 180º to let stations on the upper right and the sea ice or sea ice/ocean on the lower left?

There are two red stars in Fig. 1b. Are they same as in Fig. 1a?

We modified this illustration according to the suggestions and comments from other reviewer, including to separate Fig. 1a into two panels, enlarge the red star symbols in Fig. 1a, rotate Fig. 1b and Fig. 1c as suggestion. (Figure 1 and Lines 94-98)

L106, remove "observed" or "record"

We corrected these grammatical mistakes and inappropriate expressions according to the comments. (Line 114)

L115, Table 2, Add a column for the type of IMB (CRREL-IMB or SIMBA) deployed

We think you are referring to Table 1. We modified this table according to the suggestion. (Table 1)

L133, Label "the Russian Progress II station" on Fig. 1.

We labeled "the Russian Progress II station" in Fig. 1b according to the suggestion. (Figure 1)

L223, From Fig. 3d, the wind are dominated by easterly wind and ESE wind at ZS,

and dominated by NE, and NNE and ENE at DS station.

The wind forcing at ZS is characterized by katabatic winds, of which, winds from the east, ENE and ESE are dominant, with the frequency of 38.9%, 22.8%, and 11.7%. The wind forcing at DS is largely driven by passing synoptic systems, with the dominant wind direction from NNE to ESE, accounting 66%. We added the distribution of wind direction in Fig. 3d. (Figure 3 and Line 251)

L231, where is the Vestfold Hills? Can you label it in Fig. 1?
We labeled "Vestfold Hills" in Fig. 1d according to the suggestions. (Figure 1)

L249, remove "obtained" and replace "synchronously" with "synchronous".
We corrected these grammatical mistakes and inappropriate expressions according to the comments. (Line 254)

L265, Fig. 4, No lines for ZS2014 in Fig. 4a.
The topmost temperature thermistor of ZS2014 was just placed on the snow–ice interface at deployment as a result of an inaccurate operation. The snow depth of ZS2014 could not be retrieved from the temperature profiles and the in-situ measurements were used instead. Only nine measurements of snow depth were made during the operation period of ZS2014. In Fig.4a, these snow depth measurements are shown as blue circles, but not a line. We modified the legend of the snow depth of ZS2014 and added this information in the caption of Fig. 4. (Figure 4 and Lines 269-270)

L318, replace "; the values" with "which"
We corrected these grammatical mistakes and inappropriate expressions according to the comments. (Line 325)

L315, Fig. 6b, is it difficult to see the temperature change in S1 and S2, please change another color for them. Or add one figure for temperature gradient? This is related to your statement in L348-349.
We modified this illustration according to the suggestions by adding a subplot of temperature gradient in the current Fig. 6c. (Figure 6 and Lines 344-345)

L358, "This temporal lag …. within the ice column". Give a time for this temporal lag change".
During the event of S1 (mid July), this temporal lag was about 3 days and it reached 7 days by late August and early September (S2). We specified this change in temporal lag in Section 3.4. (Lines 359)

L365, I disagree with "the basal ice growth was primarily regulated by Fc and Fw". From Fig.7, the basal ice growth was also regulated by Fl. So please also discuss the Fl in the manuscript.
The thermal balance at the ice base assumes that the latent heat flux $F_l$ balances the thermal energy contribution, and it is calculated directly using the ice growth rate. The

definition of $F_l$ can be found in Section 2.4.

L370, replace ";, this drove" with "which drove"
We corrected these grammatical mistakes and inappropriate expressions according to the comments. (Line 378)

L402, "For a site", point out this site is which site in Table 3
L406, the same site, I think you are pointing to S1 in Heil (2006) and Heil et al. (2011). Please add this information in your text.
"For a site close to the coast near ZS", this site refers to the site closest to the coast in Section 3 and the site SIP in Table 3, and also the same site of ZS2013a, ZS2014 and ZS2015. "Similarly, at DS…from the period of 1957-2009 obtained at the same site", the same site here refers to site S1 in Heil (2006) and Heil et al. (2011). To make the expression clearer, we added this information in the Section 4.1 as suggested. (Lines 415; 421)

L429-443, Your LFI mass balance results are from the IMBs point measurements. The point measurements are more related to small-scale processes. How are you related the small-scale results to local-scale, regional-scale? Can you discuss this a little bit?
We added some discussions on the representativeness on our measurements and how to upscale the derived results. (Lines 442-447)

L448-451, Obviously your description is around DS. Could you add this information clearly in the text?
This part does describe the effect of snow layer on the LFI thickness near DS. We specified this location information. (Line 469)

L452, the largest increase in the simulated LFI thickness, as I see from Fig. 8, occurred at ZS2010(Fig. 8c), not ZS2013b (Fig. 8e). Can you re-check your results?
In Fig. 8, the y-axis ticks are different, which would be misleading that the largest increase in the simulated LFI thickness occurred at ZS2010. In fact, when the influence of snow was ignored, the largest increase in the simulated LFI thickness at ZS2010 was 0.23m, less than that at ZS2013b (0.35m). To avoid this misunderstanding, we used the same y-axis ticks in Fig. 8 as suggested. (Figure 8)

L461-462, This sentence can be rewritten as "This difference indicates that the LFI at ZS was more influenced by other factors, such as the oceanic heat flux and katabatic wind, compared at DS.
We corrected these grammatical mistakes and inappropriate expressions according to the comments. (Lines 480-482)

L470-471, Please make sure that you refer to the right Figures, Fig. 7 or Fig. 8? In Fig. 7i and 7j, one can see the larger influence of FW near ZS than near DS. But you are comparing with and without the oceanic heat flux, Fig 8 might be the right figure you

We want to illustrate the oceanic heat flux exerts a larger influence on the mass balance of the LFI near ZS than near DS, which could also be seen in Fig. 7i and 7j. We rewrote this sentence to make the expression clearer. (Lines 488-489)

L472, Make the sentence to concise. Such as rewrite as "which leads to small contributions to the oceanic heat flux to the LFI mass balance there."
To make the expression clearer, we rewrote this sentence as "which led to low value and contribution to the LFI mass balance of oceanic heat flux." (Lines 490-492)

L475-479, It seems that your increase or decrease of simulated thickness is compared to AT_obs or AT_mean. Please clarify this in your text.
To assess the effect of the oceanic heat flux on the LFI growth, we compared the evolution of the ice thickness estimated by taking into account the oceanic heat flux using Eq. (4) to those estimated ignoring this flux. To identify the impact of snow cover on the LFI mass balance from the perspective of the thermal insulation effect, we used the AT obtained from the year of observation (AT_obs) instead of Ts for the LFI thickness calculation. The forcing using AT_obs actually ignores the attenuation effect of snow cover on air temperature. We rewrote the sentence in the Section 2.5 to make the expression clearer. (Lines 215-216)

L490, using same y-axis ticks in all the subplots for Fig. 8.
We used the same y-axis ticks in Fig. 8 as suggested. (Figure 8)

L523, remove "explaining" before "why"
We corrected these grammatical mistakes and inappropriate expressions according to the comments. (Line 543)

L533, "upward shift" or "downward shift", please recheck. From Fig.2a, the snow-ice interface was downward shift.
This is a mistake in the original manuscript. We have corrected it. (Line 554)

L555, not only distribution of snow but also redistribution of snow. Please add this information in your text also.
We added the mechanism of snow redistribution in the text according to the comments. (Lines 433, 582-583)